# AuPair: Golden Example Pairs for Code Repair

**Aditi Mavalankar** [1]   **Hassan Mansoor** [1]   **Zita Marinho** [1]   **Masha Samsikova** [1]   **Tom Schaul** [1]

## Abstract

Scaling up inference-time compute has proven to be a valuable strategy in improving the performance of Large Language Models (LLMs) without fine-tuning. An important task that can benefit from additional inference-time compute is self-repair; given an initial flawed response or guess, the LLM corrects its own mistake and produces an improved response or fix. We leverage the in-context learning ability of LLMs to perform self-repair in the coding domain. The key contribution of our paper is an approach that synthesises and selects an ordered set of golden example *pairs*, or AuPairs, of these initial guesses and subsequent fixes for the corresponding problems. Each such AuPair is provided as a single in-context example at inference time to generate a repaired solution. For an inference-time compute budget of $N$ LLM calls per problem, $N$ AuPairs are used to generate $N$ repaired solutions, out of which the highest-scoring solution is the final answer. The underlying intuition is that if the LLM is given a different example of fixing an incorrect guess each time, it can subsequently generate a diverse set of repaired solutions. Our algorithm selects these AuPairs in a manner that maximises complementarity and usefulness. We demonstrate the results of our algorithm on 5 LLMs across 7 competitive programming datasets for the code repair task. Our algorithm yields a significant boost in performance compared to best-of-$N$ and self-repair, and also exhibits strong generalisation across datasets and models. Moreover, our approach shows stronger scaling with inference-time compute budget compared to baselines.

## 1. Introduction

Recent progress in the field of Large Language Models (LLMs) has resulted in models that keep getting better at generating responses to user queries. When providing these already powerful models with more inference-time compute—increasing number of LLM calls—methods that sample different responses and then select the best among them, such as best-of-$N$ (Stiennon et al., 2020) or self-consistency (Wang et al., 2023b), have shown clear benefits. While these approaches are more breadth-focused, another way to leverage inference time compute is to improve or *repair* the LLM's initial *guesses* by generating better *fixes* (Olausson et al., 2024). We propose combining the benefits of both these approaches to generate a wide set of repaired solutions for poor initial LLM responses, out of which the highest-scoring solution is the final answer.

To generate a wide range of repaired solutions for each initial LLM response, we exploit the in-context learning capability exhibited by LLMs. The main contribution of our paper is an algorithm that, given an inference-time compute budget of $N$ LLM calls, produces an ordered set of up to $N$ golden example pairs, or AuPairs[1]. Each such AuPair contains the initial guess and consequent fix for the corresponding coding problem, along with their respective unit test scores. An example AuPair is illustrated in Fig. 1. At inference time, the contents of an AuPair are concatenated as described in § A.10, and provided as a 1-shot example to generate a fix for the test problem. This is done for each of the $N$ AuPairs; of all the fixes generated, the one that gets the highest score on the unit tests is selected.

A core ingredient of our proposed algorithm involves the selection of these AuPairs. We propose a submodular approach that selects AuPairs based on the ability of each pair to solve different problems in a held-out validation set. Since the AuPairs are selected such that each subsequent AuPair solves a different set of problems than the ones solved by its predecessor AuPairs, by design, we get *complementary* AuPairs. Also, as the list of AuPairs is constructed by taking the greedy pair at each step, only those

---

[1]Google DeepMind. Correspondence to: Aditi Mavalankar <mavalankar@deepmind.com>.

*Proceedings of the $42^{nd}$ International Conference on Machine Learning*, Vancouver, Canada. PMLR 267, 2025. Copyright 2025 by the author(s).

[1]The name AuPair is a coupling of Au, the chemical symbol for gold, and Pair, jointly referring to golden pairs that are produced by our algorithm. The high-level interpretation is that like an "au pair", the approach guides the LLM towards better behaviour.

```
The Little Elephant loves numbers. He has a positive integer x. The Little Elephant
wants to find the number of positive integers d, such that d is the divisor of x, and x
and d have at least one common (the same) digit in their decimal representations. Help
the Little Elephant to find the described number.

Input
A single line contains a single integer x (1 ≤ x ≤ 10^9).

Output
In a single line print an integer — the answer to the problem.

Examples

Input
1
Output
1

Input
10
Output
2
```

```
                                   score: 0.0
def solve(x: int):              def solve(x: int):
    x = str(x)                      x = str(x)
    count = 0                       count = 0
    for i in range(1, x + 1):       for i in range(1, int(x) + 1):
        if x[0] in str(i) or str(i)[0]     if int(x) % i == 0:
in x:                                   i = str(i)
            count += 1                      for digit in x:
    print(count)                                if digit in i:
                                                    count += 1
                                                    break
                                        print(count)
                                                   score: 1.0
```

Figure 1: **An example AuPair** ⬤: The LLM-generated guess and fix, along with their respective scores for the corresponding CodeForces problem (problem description at the top). To provide this AuPair in context at inference time, the problem description, guess, and fix, are concatenated as described in Fig. 18.

pairs that lead to an increase in the fix scores are selected, resulting in *useful* AuPairs.

In this paper, we address the code repair task: given a coding problem, an initial guess which is LLM-generated code, and a set of test cases that are used only to evaluate the correctness of the generated code, the LLM has to generate an improved fix for the problem. We show that the fixes generated by AuPair are significantly more *useful* and *diverse* than those generated using best-of-$N$ (§3) for the same inference-time compute budget. We also show that AuPair outperforms self-repair (Olausson et al., 2024), which generates intermediate verbal feedback before generating repaired solutions for a problem. The key contributions of our paper are the following:

- An inference-time **algorithm**, which constructs a golden set of code repair examples, or AuPairs, that boost performance significantly when used as in-context examples (§2).

- **Reliably outperforming** best-of-$N$ (Stiennon et al., 2020) and self-repair (Olausson et al., 2024) across 5 different models: Gemini-1.5-Pro, GPT-4o-mini, Gemini-1.5-Flash, Gemma-27B, Gemma-9B (§3.1).

- Strong **scaling** performance with inference time compute, with far less diminishing returns than best-of-$N$ and self-repair (§3.3).

- Robust out-of-distribution **generalisation**, across both model sizes and datasets (§3.4).

## 2. Approach

The goal of our proposed approach in the coding domain is to improve code repair performance on unit tests at inference time by curating a list of pairs that can be provided as in context examples. The code repair prompt includes an in-context example, followed by a text description of the problem to solve and the initial *guess* generated by the LLM. The LLM then generates a revision, or a *fix* that improves performance on the unit tests for that problem. In the prompt, we include the scores achieved by the guess and fix on the unit tests, but no other execution feedback.[2]

In order to disentangle repair performance from the quality of initial guesses, we first curate composite datasets consisting of initial guesses for all the coding problems. Given a dataset consisting of problems and their corresponding tests, we generate an initial *guess* for each problem and compute its score on the unit tests. If the guess passes all the unit tests for that problem correctly, no further improvement is required and we discard that problem. If not, we add this guess along with its corresponding score and problem as a datapoint to our curated dataset. This dataset is then divided into training, validation, and test datasets. We use the training dataset $\mathcal{D}_{\text{train}} \equiv \mathcal{D}$ for pair generation (Fig. 2), and the validation dataset $\mathcal{D}_{\text{val}}$ for AuPair extraction. The test dataset $\mathcal{D}_{\text{test}}$ is used in the final testing phase.

Following the creation of these three datasets, we now discuss our approach, which consists of two main phases: 1) Pair Generation (§2.1), and 2) AuPair Extraction (§2.2).

### 2.1. Phase 1: Pair Generation

In this phase, we generate a large set $\mathcal{C}$ of pairs that are potential candidates for our final set of AuPairs. This is done in the following manner: a problem along with its initial guess is sampled from the training dataset $\mathcal{D}$. The LLM generates a fix for this guess. If this generated fix has a higher score on the unit tests for that problem than the initial guess, this guess-fix pair is added to $\mathcal{C}$. Furthermore, if this fix is imperfect, i.e. it does not pass all the unit tests, it becomes a potential guess with further scope for improvement, so it is added as a new guess to the training dataset $\mathcal{D}$. This is repeated several times to collect a large set of such candidate pairs.[3] A visual illustration of this phase is provided in Fig. 2. While we include the pair generation phase for completeness, it is important to note that in several

---

[2]The repair prompt is composed using the prompting strategy shown in Fig. 18.

[3]Since the aim is to collect a large set of pairs, we want the LLM to generate a wide variety of fixes. For this, we compose a $k$-shot prompt for repair, in which the $k$ in-context example pairs are randomly sampled from the existing set $\mathcal{C}$ for each sampled problem. As the LLM generates more fixes, $\mathcal{C}$ gets populated, subsequently resulting in more diverse prompts.

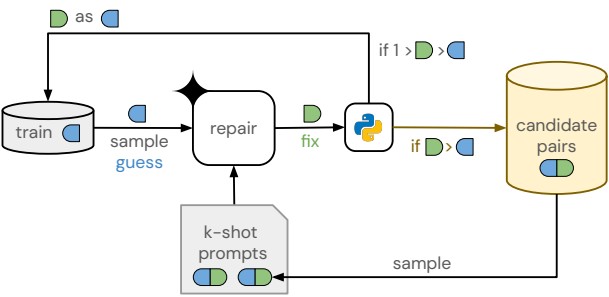

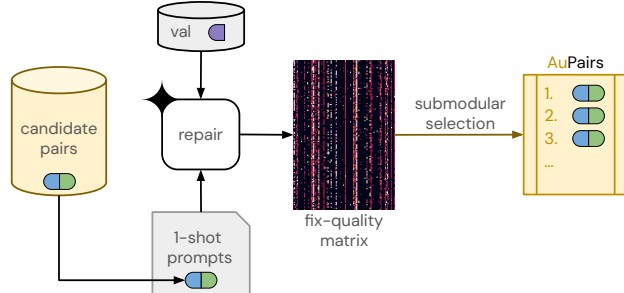

Figure 2: **Pair Generation:** We collect a large set of guesses for coding problems and their fixes yielding candidate pairs that will later be used to get AuPairs. A problem with its guess is sampled from the training dataset, with $k$ randomly sampled few-shot pairs from the candidate pair buffer. This $k$-shot prompt is passed through an LLM to generate a fix, which is evaluated on the unit tests. If this fix is better than the guess, this (guess, fix) pair is added to the set of candidate pairs. Imperfect fixes are added as new guesses to the training dataset (See §2.1).

Figure 3: **AuPair Extraction:** Given a large set of candidate pairs, the next step is to extract AuPairs. Each pair is provided as a 1-shot in-context example in the prompt for each problem and its guess from the validation dataset. These prompts are passed to the LLM which generates fixes that are evaluated on unit tests to populate a fix-quality matrix, as described in Algorithm 1. Next, a submodular selection mechanism is applied on this fix-quality matrix to obtain the list of AuPairs (Algorithm 2).

other domains, paired data may already be available. In such cases, the set of candidate pairs $\mathcal{C}$ that we curate in this step, can simply be replaced by the given paired data.

---

**Algorithm 1** Fix quality matrix computation

**Require:** $\begin{cases} \text{LLM} & \text{large language model} \\ \mathcal{C} & \text{candidate pairs} \\ \mathcal{D}_{\text{val}} & \text{validation dataset} \\ \text{score} & \text{code eval function} \end{cases}$

1: init fix quality matrix $M \leftarrow \mathbf{0}^{|\mathcal{C}| \times |\mathcal{D}_{\text{val}}|}$
2: **for** pair $c_i$, problem $x_j \in \mathcal{C} \times \mathcal{D}_{\text{val}}$ **do**
3:      build 1-shot prompt: $p \leftarrow c_i \parallel x_j$
4:      generate fix: $\hat{y} \leftarrow \text{LLM}(p)$
5:      evaluate fix: $M_{i,j} \leftarrow \text{score}(\hat{y})$
6: **end for**
7: **return** $M$

---

### 2.2. Phase 2: AuPair Extraction

Now that we have a large set $\mathcal{C}$ of candidate pairs, the next step is to determine which of these will actually help boost performance, i.e., which of these are AuPairs. We do this in a *submodular* fashion by making use of the validation dataset $\mathcal{D}_{\text{val}}$. For each pair-problem combination $(c_i, x_j) \in \mathcal{C} \times \mathcal{D}_{\text{val}}$, we build a 1-shot prompt $p$ using the prompting strategy described in Fig. 18. This 1-shot prompt $p$ is given as input to the LLM, which generates a fix for the given problem $x_j$. The fix generated by the LLM is then evaluated on the unit tests and stored in the fix quality matrix $M \in \mathbb{R}^{|\mathcal{C}| \times |\mathcal{D}_{\text{val}}|}$ at index $(i, j)$. This part of AuPair extraction is outlined in **Algorithm 1**.

Next, we use this fix quality matrix $M$ to extract the AuPairs by taking the following steps: 1) Select the pair that gets the highest mean score across all problems in $\mathcal{D}_{\text{val}}$, say $c_k$, and add it to the list of AuPairs $\mathcal{A} : \mathcal{A} \leftarrow \mathcal{A} \cup c_k$. This is a greedy way of selecting the best pair given all previous AuPairs and produces an ordered set of AuPairs. 2) Subtract the row score $M_k$ (i.e. score on all the problems in $\mathcal{D}_{\text{val}}$) of this newly added pair from all the rows in the fix quality matrix with an update: $M - M_k$. This ensures that redundant AuPairs are not produced by the approach. The updated fix quality matrix is clipped to $(0, 1)$ since any negative value in the matrix $M$, say $M_{i,j}$, implies that the problem $x_j$ cannot be improved further by pair $c_i$. Without clipping, we would not get an accurate estimate of the improvement in the next step of submodular extraction. 3) Repeat this process until improvement falls beyond a certain tolerance $\epsilon$.

---

**Algorithm 2** Submodular AuPair extraction

**Require:** $\begin{cases} M & \text{fix quality matrix} \\ \mathcal{C} & \text{candidate pairs} \\ \epsilon & \text{tolerance} \end{cases}$

1: initialise AuPairs $\mathcal{A} \leftarrow []$
2: **repeat**
3:      per-pair scores: $\bar{m} \leftarrow \text{row-mean}(M)$
4:      get best pair: $c_k \leftarrow \text{argmax}_{\mathcal{C}} \bar{m}$
5:      append to AuPairs: $\mathcal{A} \leftarrow \mathcal{A} + c_k$
6:      update $M \leftarrow \text{clip}(M - M_k, 0, 1)$
7: **until** $\max(\bar{m}) < \epsilon$
     **return** $\mathcal{A}$

---

This process of iteratively constructing the set of AuPairs ensures that they improve performance on disjoint parts of

the problem space. The AuPairs that we obtain from this phase are used in the same manner at inference time, as 1-shot examples, to improve code repair performance. The compute budget $N$ determines the number of AuPairs that we can use at inference time. Since the AuPairs form an ordered set, the first $N$ AuPairs are used for a compute budget of $N$ LLM calls. The final solution for each problem is the one that passes the most test cases, among all generated solutions. This submodular extraction of AuPairs is shown in **Algorithm 2**. Fig. 3 has a joint diagram depicting fix quality matrix computation and submodular AuPair extraction.

## 3. Experiments

**Datasets:** We use 7 datasets that contain problems and test cases from competitive programming contests: 1) Code-Forces (8.8k problems), 2) AtCoder (1.3k problems), 3) HackerEarth (1.2k problems), 4) CodeChef (768 problems), 5) LiveCodeBench (400 problems), 6) CodeJam (180 problems), and 7) Aizu (2.2k problems) (Li et al., 2022b; Jain et al., 2024). We choose CodeForces and AtCoder, separately, for in-distribution testing, and use the rest exclusively for out-of-distribution testing. Our training / validation / test split proportions for the CodeForces and AtCoder datasets are 37.5/12.5/50%. Some datasets have difficulty levels as part of the problem; for those we maintain the same stratified distribution of questions in the training, validation and test datasets.

**Models:** We demonstrate the superior code repair capability of AuPair on 5 different models: Gemini-1.5-Pro, GPT-4o-mini, Gemini-1.5-Flash, Gemma-27B, and Gemma-9B. In addition to using these models for dataset curation and pair generation, we also look at the transfer capabilities of our method with respect to different models in Section 3.5.

**Evaluation:** We conduct two types of evaluation: in-distribution and out-of-distribution. For in-distribution evaluation, we use the test split from the same dataset as the one used for pair generation and AuPair extraction. This ensures that the format of questions and test cases in the test questions matches that of the AuPairs. Out-of-distribution evaluation uses a different coding dataset; this means that the test samples have different format of questions, difficulty, types of problems and test cases than the AuPairs. Another axis of out-of-distribution evaluation is the model axis: we report performance obtained using AuPairs produced by a different model than the one used for inference.

**Metrics:** Our primary metric is the commonly used test pass rate, also called test case average (Hendrycks et al., 2021; Ouyang et al., 2024; Wu & Fard, 2024), which we compute as the average percentage of test cases passed, as done in prior work (Tang et al., 2024; Ding et al., 2024; Chen et al., 2023). In our setting, since we choose the best

out of $N$ responses generated by the LLM, the test pass rate for a test dataset with $P$ problems is calculated as:

$$
\frac{1}{P} \sum_{p=1}^{P} \max_{i \in \{1,\ldots,N\}} \frac{1}{|T_p|} \sum_{j=1}^{|T_p|} \mathbb{1}\{\texttt{eval}(\text{code}_{p,i}, T_{p,j}) == 1\}
\tag{1}
$$

where $T_p$ refers to the unit tests for problem $p$, and $\text{code}_{p,i}$ is the code generated by the LLM for problem $p$ in the $i^{\text{th}}$ LLM call. The innermost loop computes the percentage of unit tests passed by the LLM output $\text{code}_{p,i}$. Next, we select the code output that has the highest value for percentage of unit tests passed, indicated by the max operation over all LLM calls $i \in \{1, \ldots, N\}$. The outermost loop averages this across all the problems in the test dataset.

We also report the results of strict accuracy (Hendrycks et al., 2021), which is the percentage of generated solutions that pass all test cases (definition and results in Appendix §A.2). Note that we cannot provide fair results for the pass@$k$ metric because that involves making an assumption that all $k$ responses from the LLM are i.i.d. generated, whereas our approach produces fixes in a specific order. In our case, the first AuPair is more useful than the second, which is more useful than the third, etc; so their success probabilities monotonically decrease as a function of $k$.

**Baselines:** We compare the effectiveness of our proposed approach with best-of-$N$ (Stiennon et al., 2020) and self-repair (Olausson et al., 2024). Best-of-$N$ is a strong baseline to improve model performance by allowing multiple LLM calls at inference time. Of the $N$ generated responses, the highest-scoring response is selected. To ensure the sampling of high-quality diverse responses in best-of-$N$, we set the temperature to 1.0 (Renze & Guven, 2024). Self-repair, on the other hand, uses the compute budget of $N$ LLM calls to either generate verbal feedback or repaired code. Our compute budget is $N = 32$, of which 4 LLM calls are used to generate verbal feedback and 7 LLM calls to generate repaired code for each verbal feedback.

The remainder of this section will discuss a plethora of empirical results, on overall and ablated performance (§3.1 and 3.2), scalability and generalisation (§3.3 to 3.6), and diversity (§3.7 to 3.10).

### 3.1. Significantly Boosted Code Repair Performance

The first step to assess code repair performance is to measure *in-distribution* performance; namely generating and selecting AuPairs on the training and validation sets that match the test dataset, and using the same model for evaluation as AuPair construction. We do this for 2 datasets (Code-Forces and AtCoder) and all 5 models. Fig. 4 shows the

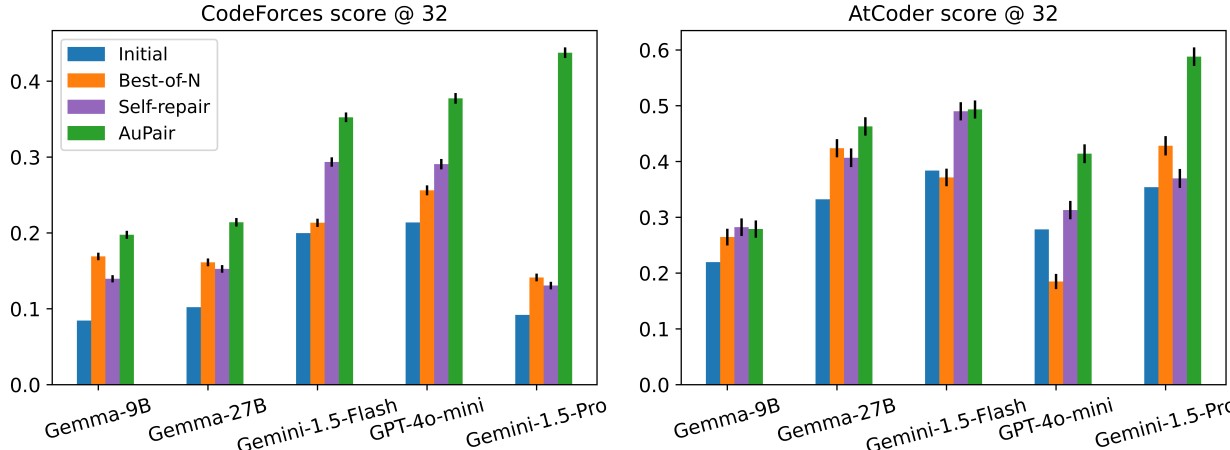

Figure 4: **In-distribution code repair performance:** with $N = 32$ LLM calls at inference time and the same train / val / test data distribution, we compute the test pass rate. The same model is used for generating the initial guesses and fixes and the AuPair extraction. CodeForces (left, 8.8k problems) and AtCoder (right, 1.3k problems), see §3.1 for more details.

resulting comparison between the best-of-$N$ and self-repair baselines and AuPair, for a budget of $N = 32$ LLM calls at inference time.[4] AuPair is clearly superior to best-of-$N$ and self-repair (matching in a few cases) on all models and datasets, sometimes by wide margins. This clearly establishes that our proposal of providing a different in-context example of code repair in each LLM call can significantly boost performance.

An interesting side-result is visible in initial performance, i.e., the performance of the initial guesses, which have to be repaired. Gemini-1.5-Pro, despite being a superior model to Gemini-1.5-Flash, shows worse initial performance. Since the code generated has certain conditions that allow successful execution, we observe that many initial guesses of generated code fail because they do not obey these conditions (see Appendix §A.11). In such cases, code repair with either best-of-$N$ or self-repair is unlikely to give us high boost in performance since the initial solution is badly formatted. This is one clear case where having an AuPair in context significantly improves performance. Thus, using AuPairs in conjunction with high performing models leads to large performance improvements despite poor initial performance, as we can see for both CodeForces and AtCoder with the Gemini-1.5-Pro model in Fig. 4. This also mitigates the need for more sophisticated prompt engineering to a large extent.

---

[4]Since our algorithm yields a variable number of AuPairs, for datasets with fewer generated pairs, this number can be less than 32. To have a fair comparison, we set the same compute budget $N$ for best-of-$N$ and self-repair. This is the case for AtCoder (Fig. 4, right), where our algorithm yields 14, 15, and 27 AuPairs for Gemma-9B, GPT-4o-mini, and Gemma-27B respectively. So the corresponding baselines also use the same compute budget.

### 3.2. Selection Matters: AuPairs are More Effective than Random Pairs

We design an ablation to disentangle the two possible sources of improvement that our approach demonstrates, namely 1) in-context learning and 2) the choice of AuPairs. It is not implausible for the boost in performance to result from the LLMs' in-context learning ability, and that the same result could be achieved by including *any* set of pairs. On the other hand, our approach specifically targets complementarity during construction of AuPairs in that subsequent AuPairs are selected based on their ability to solve problems that previous AuPairs were unable to solve. To resolve this, we compare the full method to a random-pair baseline that randomly selects pairs from the full candidate set (the result of Phase 1), deduplicating the problems that the random pairs solve (which makes it a stronger baseline). Fig. 5(a) shows that AuPair significantly outperforms the random-pair baseline for $N = 1, ..., 32$, saving $2.5 - 3\times$ more compute by achieving the same score with 12 AuPairs as the random pair baseline gets with 32 AuPairs. Note that for any fixed candidate set, as $N$ grows toward the size of the full set of pairs, the performance of the random-pair baseline will equal that of AuPair.

### 3.3. Better Scaling with Inference-Time Compute

At a fixed budget of $N = 32$ LLM calls, our results look promising. In this section, we investigate whether and how performance scales with the compute budget $N$. Fig. 5(b) plots the score as a function of $N$ using Gemini-1.5-Pro on the CodeForces dataset (additional scaling curves for Gemini-1.5-Flash and strict accuracy metric in the Appendix, see Figs. 10 and 13). For each additional LLM call, we use the next best AuPair produced by the algorithm

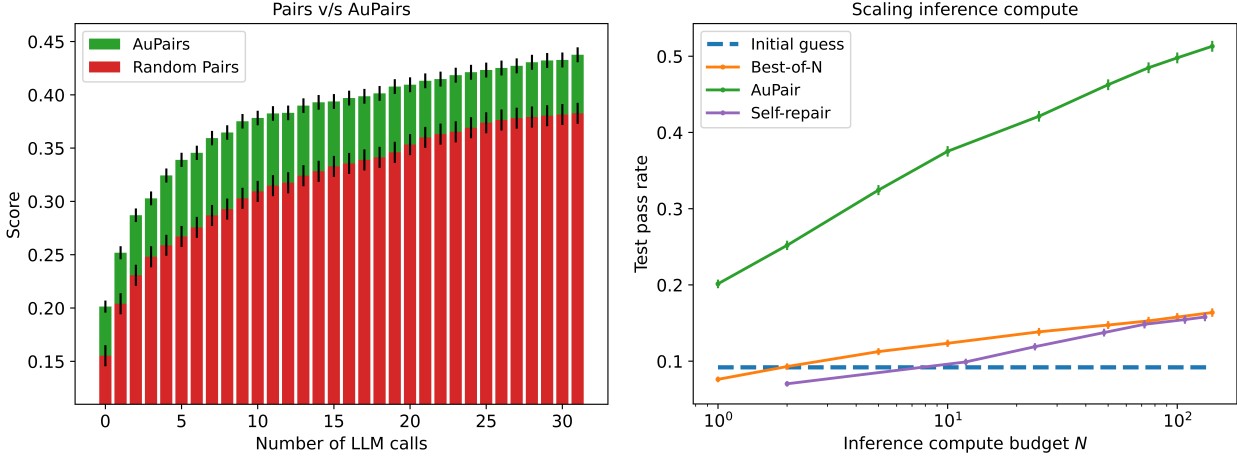

Figure 5: **(a) AuPairs vs. random pairs**: AuPairs (green) are far ($\sim 2.5 - 3\times$) more compute-efficient than random pairs (red); it takes only 12 AuPairs to reach the same performance as 32 random pairs; **(b) Scaling inference-time compute:** the score increases with compute budget at a much steeper rate using AuPairs (CodeForces, Gemini-1.5-Pro).

and provide it in context to generate the LLM response. Our algorithm produces 144 AuPairs for the CodeForces dataset using Gemini-1.5-Pro, and achieves a test pass rate of 51.32% and strict accuracy of 39.73% (see §A.3) at 144 LLM calls. The results shows a clear scaling trend with a consistent log-linear performance increase as a function of compute, without any sign of a plateau. More importantly, the increase is substantially *steeper* than best-of-$N$ and self-repair (which achieve test pass rate of 16.05% and 15.79% and strict accuracy 12.04% and 12.23% respectively); in other words, prompting with in-context complementary AuPairs makes more efficient use of compute than either repeated sampling given a fixed repair prompt, or repair with model-generated verbal feedback.

### 3.4. Strong Generalisation to Out-of-distribution Datasets

The aim of this experiment is to determine whether our approach exhibits out-of-distribution generalisation, i.e., given AuPairs collected on a different dataset, see if we can retain the performance improvements that we obtain in-distribution. To test this, we evaluate the AuPairs collected using Gemini-1.5-Pro on CodeForces on the other 6 datasets and compare them with the corresponding baselines. Fig. 6 shows that across datasets, our approach outperforms baselines by a large margin, despite having out-of-distribution AuPairs. This implies that the process of collecting AuPairs may only be needed on one dataset and the benefits can be reaped across a wide range of problems (from other datasets, or users) at inference time.

### 3.5. Decent Cross-Model Transfer

Now that we have seen that our approach can exhibit very good out-of-distribution generalisation along the data axis, we evaluate it on its ability to generalise on the model axis, i.e., we look at the performance of AuPairs collected using a different model. We evaluate this cross-model transfer capability for several model combinations on CodeForces. The resulting 16 ablations are shown in Fig. 7(a), and help disentangle the impact of the AuPairs versus the code repair capabilities of the inference model. A key takeaway is that the Gemma models exhibit worse performance, regardless of the quality of AuPairs used at inference time, indicating that they are inferior at the capability of code repair. Gemini-1.5-Flash performs much better at code repair, and its sensitivity to the source of AuPairs is negligible: it is equally performant for each source. Gemini-1.5-Pro, on the other hand, *is* sensitive to the source of AuPairs; in particular, when Gemini-1.5-Pro uses AuPairs collected by the same model, it achieves the best performance by a large margin. With AuPairs selected using other models, Gemini-1.5-Pro achieves comparable performance to Gemini-1.5-Flash. One reason for the standout performance when using Gemini-1.5-Pro AuPairs seems that those examples result in substantially more diverse generations, as shown in Section 3.7. However, Fig. 7(a) as a whole suggests that there is an ordering in terms of performance: 1) the model used at inference time has to have good code repair capabilities, and 2) the stronger the model is at code repair, the more improvement we can expect from it with a higher *quality* of AuPairs. We also show additional results for cross-model transfer on GPT-4o in §A.4.

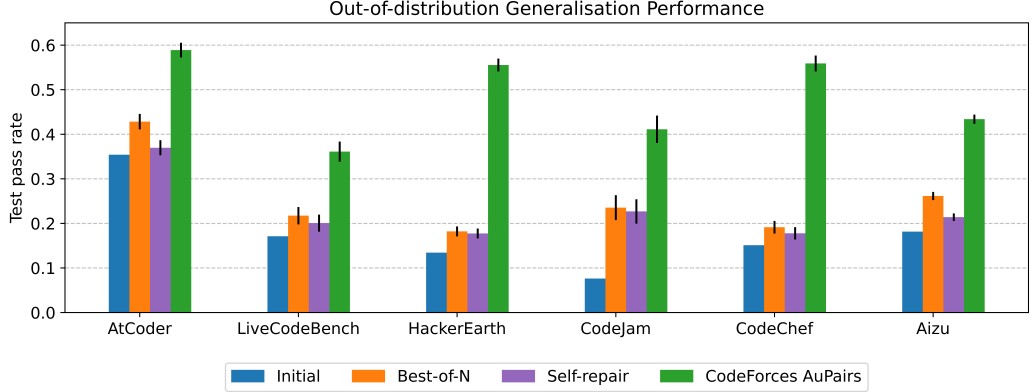

Figure 6: **Out-of-distribution code repair performance:** AuPairs extracted on the CodeForces dataset show strong generalisation performance across the other six datasets (model: Gemini-1.5-Pro, metric: test pass rate).

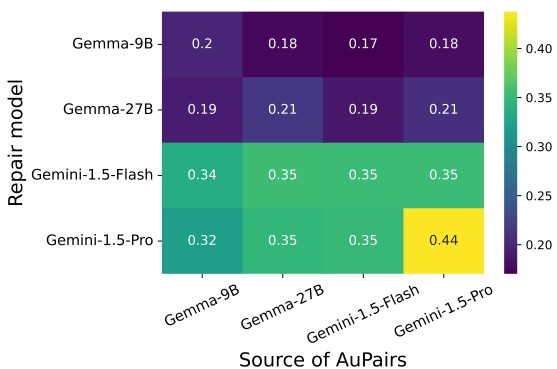

Figure 7: **Cross-model transfer:** AuPair shows good cross-model transfer capabilities on CodeForces.

### 3.6. Better scaling with validation data

We conduct experiments with smaller validation sets to curate AuPairs and report the results in Table 1. The takeaway is that the larger the validation set, the more distinct complementary improvements can be observed, and hence the larger the maximal set of AuPairs that can be discovered. So, larger validation sets make it possible to effectively scale up to more inference compute. However, even just looking at the top 32 AuPairs (which is a fair comparison for varying validation set sizes), we find that their quality increases monotonically with the size of the validation set.

### 3.7. High Code-specific Diversity

We use Abstract Syntax Trees (ASTs) to study the fixes generated using AuPairs. Since there are $N$ fixes for each problem ($N = 32$), we measure the diversity per problem as the number of unique changes made to the guess over all $N$ fixes for that problem. The diversity score is calculated as the average number of unique ASTs generated per problem. We perform the set difference of all subtrees in the fix AST

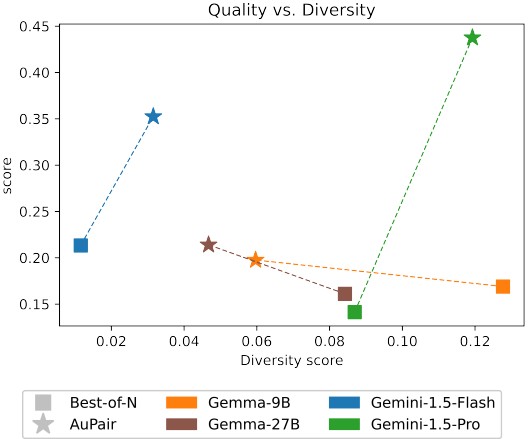

Figure 8: **Diversity-Score plot:** AuPair ($\star$) generates more diverse responses than best-of-$N$ ($\square$) for Gemini-1.5-Flash and -Pro; this trend is reversed for Gemma models. AuPair always generates higher-scoring fixes than best-of-$N$.

and the guess AST, and normalise by the maximum number of subtrees. We plot this diversity metric against the score to gauge how diverse and useful the AuPairs are, compared to best-of-$N$, in Fig. 8. §A.9 contains details on the diversity score computation. The results show that while AuPairs always increase performance, they yield more diverse fixes when given to the more competent models (Gemini-1.5-Pro and -Flash), and less diverse fixes for the Gemma models. The superior performance of AuPairs produced by Gemini-1.5-Pro corresponds to highly diverse fixes (Fig. 8, top right).

### 3.8. High Repair Diversity

In addition to code diversity, we also analyse the repair diversity of the solutions generated by AuPair compared to the baselines. In this, we report the following values: 1) percentage of problems in which the code was reformatted

| Validation set size | # of AuPairs | Test pass rate |
|---|---|---|
| Random | N/A | 0.383 |
| 10% | 32 | 0.403 |
| 25% | 52 | 0.418 |
| 100% | 144 | 0.438 |

Table 1: **Scaling with validation data:** We see an increase in 1) the number of AuPairs discovered, and 2) performance using the top 32 AuPairs on scaling validation data.

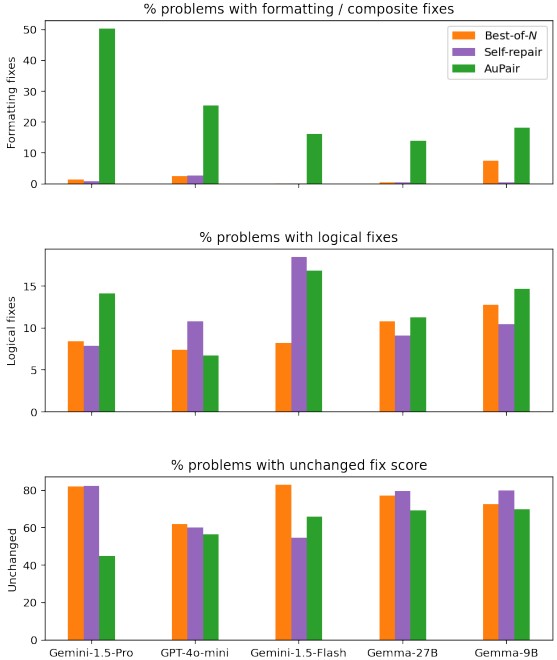

Figure 9: **Repair Diversity across models and approaches:** In most cases, AuPair outperforms baselines by performing a diverse set of fixes like formatting, composite, logical fixes. This diversity is also reflected in the lower number of problems where the fix is equivalent to the guess.

to increase the score by obeying the formatting constraints described before (higher is better), 2) percentage of problems in which the fix generated an improved solution by just changing the logic (higher is better), and 3) percentage of problems in which the fix score remained unchanged with respect to the guess (lower is better). Note that fixes that involve multiple changes, such as formatting changes and logical changes, are counted under formatting fixes. Fig. 9 provides a visual illustration of these results.

### 3.9. Improvement on All Difficulty Levels

Coding datasets have heterogeneous difficulty. We conduct additional analysis to determine *which problem levels* are most helped by AuPair, compared to the quality of initial guesses. Table 2 shows the absolute improvement in

test pass rate, i.e., the increase in this score achieved by AuPair on CodeForces. The two key observations are (a) AuPair helps significantly at all difficulty levels across models, and (b) there are larger improvements on easier levels; this trend is consistent across models. Note that the initial performance of Gemini-1.5-Pro is low because the initial guesses do not adhere to the instruction (elaborated in Appendix §A.11); however since this is the strongest model and has the best overall performance, the increases in score are significantly higher than other models.

### 3.10. Using AuPairs may lead to worse responses

While AuPairs have been shown to significantly boost performance in a best-of-$N$ setting, they can occasionally generate a worse repaired response compared to the initial response in some cases. Table 3 contains the percentage of CodeForces problems in which some fixes were worse than their initial guesses. Note that the performance gains shown earlier still hold, since for measuring performance, the *best* scoring response is selected. As we can see from Table 3, in most cases, using AuPair results in an increase in the number of problems for which a fix is worse than the initial guess. This is to be expected since AuPair is an algorithm that in addition to boosting performance also boosts diversity of the generated responses.

## 4. Related Work

Automatic Program Repair (APR) has been a longstanding area in machine learning (Devlin et al., 2017; Bhatia & Singh, 2016; Chen et al., 2019; Feng et al., 2020; Berabi et al., 2021; Chakraborty et al., 2022; Yuan et al., 2022). Most methods rely on supervised fine-tuning (SFT) to adapt LLMs to the task of code generation using labeled pairs of broken / fixed code pairs, which is costly to obtain and often task- and problem-specific (Hu et al., 2022; Jiang et al., 2021; Xia & Zhang, 2022; Dinella et al., 2020). On the other hand, unsupervised APR is challenging since it requires syntactic and semantic understanding of code, and most automatic code breaking approaches tend to be out-of-distribution with real samples. Yasunaga & Liang (2021) train a breaker and a fixer to learn to propose new code fixes that are realistic, and use a compiler to verify correctness. Our work uses model-generated partial or complete fixes on initial broken code to perform repair.

More recently, a few unsupervised approaches have been proposed based on the capability of LLMs to generate code (Chen et al., 2021; Nijkamp et al., 2023; Chowdhery et al., 2024; Li et al., 2022a; Fried et al., 2023; Li et al., 2023). APR still remains challenging, even though models are better at generating code (Olausson et al., 2024; Chen et al., 2023). Zhao et al. (2024) use a step-by-step method to repair code using a reward model as a critic, providing

| Difficulty level → | A (671) | B (675) | C (671) | D (666) | E (649) | F+ (537) |
|---|---|---|---|---|---|---|
| Gemma-9B | 0.34 (+0.16) | 0.23 (+0.13) | 0.19 (+0.12) | 0.15 (+0.09) | 0.14 (+0.08) | 0.12 (+0.07) |
| Gemma-27B | 0.28 (+0.1) | 0.25 (+0.12) | 0.20 (+0.12) | 0.19 (+0.1) | 0.17 (+0.1) | 0.20 (+0.11) |
| Gemini-1.5-Flash | 0.54 (+0.2) | 0.39 (+0.18) | 0.34 (+0.15) | 0.18 (+0.11) | 0.26 (+0.12) | 0.28 (+0.11) |
| Gemini-1.5-Pro | 0.62 (+0.42) | 0.52 (+0.4) | 0.43 (+0.35) | 0.38 (+0.32) | 0.32 (+0.28) | 0.35 (+0.29) |

Table 2: **Difficulty-wise analysis:** test pass rate using AuPairs, categorised by difficulty level from easy (A) to hard (F+), accompanied by number of problems, absolute improvement in parentheses.

| Model | Approach | % problems |
|---|---|---|
| Gemini-1.5-Pro | Best-of-$N$ | 10.52 |
|  | Self-repair | 7.62 |
|  | AuPair | 11.63 |
| GPT-4o-mini | Best-of-$N$ | 20.09 |
|  | Self-repair | 11.87 |
|  | AuPair | 15.28 |
| Gemini-1.5-Flash | Best-of-$N$ | 9.47 |
|  | Self-repair | 22.28 |
|  | AuPair | 11.79 |
| Gemma-27B | Best-of-$N$ | 14.86 |
|  | Self-repair | 9.72 |
|  | AuPair | 15.21 |
| Gemma-9B | Best-of-$N$ | 13.16 |
|  | Self-repair | 9.38 |
|  | AuPair | 13.09 |

Table 3: **Failure analysis:** increase in the % of problems in which using AuPairs resulted in a lower-scoring fix.

feedback to finetune an LLM. Shypula et al. (2024) propose a retrieval based few-shot prompting approach with Chain-of-Thought (CoT) reasoning traces, and use SFT to finetune a model using self-play. The main disadvantage of using SFT comes from the need to finetune the model to the task, which becomes more costly with increasing model sizes. In recent years, in-context learning (ICL) (Brown et al., 2020) has emerged as a flexible and compute-efficient adaptation approach to new tasks (Von Oswald et al., 2023; Akyürek et al., 2023). Le et al. (2022) use an LLM to generate code and a critic to predict functional correctness of the generated program with zero-shot transfer to new tasks. Gou et al. (2024) employ tool use to provide feedback for the LLM to self-correct via additional calls to evaluate its own output. Wang et al. (2023a) also make use of external tools and use an LLM in a learner / teacher role to provide a chain of repairs to fix the code.

Yin et al. (2024) propose an automated self-repair approach with few-shot prompting using CoT and execution feedback. Agarwal et al. (2024) also use CoT rationales but remove them from context when using few-shot prompting. Olausson et al. (2024) show that using an LLM as a feedback source for self repair has limitations when compared with independent model calls for the same problem since the ability to generated better code may be connected to the ability to identify its faulty behaviour. Welleck et al. (2023) decouple generation and correction by independently training a corrector with scalar and natural language feedback to correct intermediate imperfect generations. Ding et al. (2024) teach code-competent models to self-refine by factoring in the execution results from previous incorrect code and updating the pretrained model using this feedback. Tang et al. (2024) provide a stochastic approach for choosing which code node to expand by using a Thompson sampling approach; they leverage an exploitation-exploration mechanism for evolving the tree of code refinements.

Yuan & Banzhaf (2017) propose a multi-objective evolutionary algorithm to search over possible correct code patches; Romera-Paredes et al. (2023) use an island-based evolutionary method to encourage exploration of diverse programs, and perform iterative best-shot prompting to improve the quality of the generated code. We use a generative approach; closer to the work of Shirafuji et al. (2023), we make use of ICL abilities of LLMs to generate improved code repairs, but we provide an extra submodular process to select the samples, encouraging diversity.

## 5. Conclusions and Future Work

We propose an algorithm, AuPair, which produces an ordered set of golden example pairs each of which can be provided as an in-context example using 1-shot prompting with an inference compute budget of $N$ LLM calls to improve code repair performance at inference time. AuPair is highly scalable, showing significantly better outcomes than best-of-$N$ and self-repair, both of which are known to improve performance as inference compute is scaled up. In addition to this, the AuPairs generated using our algorithm show strong out-of-distribution generalisation and thus can be reused at inference time to solve a wide range of problems. While in this paper we have explored repair in the coding domain, future work can look at using it in other settings in which an initial solution generated by an LLM can be improved via repair. Additionally, the choice of coding implies that all our feedback is grounded, but using ungrounded feedback from reward models to build the AuPairs might be another direction worth exploring.

## Acknowledgements

The authors would like to thank Dan Calian for valuable feedback and suggestions on the paper; David Silver for his sponsorship; Wojtek Czarnecki, Kate Baumli, Jakub Sygnowski, Victor Cărbune, Volodymyr Mnih, Mina Khan, Georg Ostrovski, Shantanu Thakoor, Lei Zhang, Disha Shrivastava, Feryal Behbahani, the RL team, and the wider DeepMind community for helpful discussions.

## Impact Statement

The goal of our paper is to advance capabilities of LLMs in the domain of code repair. This work may have potential societal consequences, which we do not feel the need to highlight here.

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

# A. Appendix

## A.1. Pair Generation

In this section, we discuss the specifics of the pair generation phase and provide results pertaining to this phase. The approach that we use for pair generation is provided in Algorithm 3. Note that this is one way to generate pairs; they can be generated in other ways, or be available beforehand. Studying the impact of using pre-generated pairs for extracting AuPairs could be an interesting avenue for future work.

We set $k = 32$ in this algorithm. The reason for this is that during pair generation, we want diverse pairs to be generated, and using a different set of $k$ examples with the same problem could give us different fixes.

For the AtCoder dataset, we set a budget of 10,000 LLM calls for pair generation. Since the CodeForces dataset is larger, we set a budget of 35,000 LLM calls to maintain a good balance between having enough LLM calls per problem and maintaining the affordability of the overall approach in terms of computational resources. We report the number of pairs generated on both of these datasets across all 5 models: Gemini-1.5-Pro, GPT-4o-mini, Gemini-1.5-Flash, Gemma-27B, and Gemma-9B in Table 4. Here we provide some additional results that we were unable to include in the main text.

| CodeForces | # Pairs | # AuPairs |
|---|---|---|
| Gemini-1.5-Pro | 1560 | 144 |
| GPT-4o-mini | 1192 | 94 |
| Gemini-1.5-Flash | 1327 | 110 |
| Gemma-27B | 509 | 77 |
| Gemma-9B | 556 | 122 |

| AtCoder | # Pairs | # AuPairs |
|---|---|---|
| Gemini-1.5-Pro | 927 | 64 |
| GPT-4o-mini | 378 | 15 |
| Gemini-1.5-Flash | 397 | 64 |
| Gemma-27B | 295 | 27 |
| Gemma-9B | 147 | 14 |

Table 4: Number of pairs collected during phase 1 of the algorithm (# of pairs) and number of AuPairs extracted in phase 2 (# AuPairs): CodeForces (left) and AtCoder (right) for all 5 models.

## A.2. Measuring correctness in terms of solved problems

In addition to pass rate of unit tests, we also report the percentage of fully solved problems, for which the generated code passes all test cases. This is the *strict accuracy* metric:

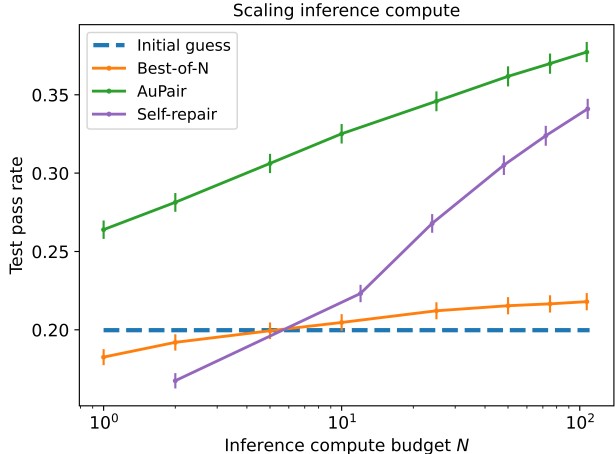

Figure 10: Scaling up inference compute on the CodeForces dataset with Gemini-1.5-Flash. Scores correspond to average pass test rate on all the test problems.

$$\frac{1}{P} \sum_{p=1}^{P} \max_{i \in \{1,\ldots,N\}} \prod_{j=1}^{|T_p|} \mathbb{1}\{\texttt{eval}(\texttt{code}_{p,i}, T_{p,j}) == \texttt{pass}\}$$

where $T_p$ refers to the unit tests for problem $p$, and $\texttt{code}_{p,i}$ is the code generated by the LLM for problem $p$ in the $i^{\text{th}}$ LLM call. Here, the innermost loop, like test pass rate, computes the percentage of unit tests passed by the LLM output $\texttt{code}_{p,i}$. Following this, we select the code output that passes all tests (max over binary values yields 1 if any such output exists, otherwise 0). The outermost loop averages this across all the problems in the test dataset.

We see that AuPair outperforms all other baselines on all models across the board, with results for CodeForces and AtCoder shown in Fig. 11.

We also show the results for out-of-distribution generalisation on this strict accuracy metric in Fig. 12; again, the results clearly indicate that AuPair outperforms all baselines on this metric as well across all datasets.

## A.3. Scaling Inference Compute

In addition to the scaling experiment we performed using Gemini-1.5-Pro (results in Fig. 5(b)), we also perform the same scaling experiment using Gemini-1.5-Flash and show the results in Fig. 10. Moreover, we report the results of the same scaling experiment on the strict accuracy metric in Fig. 13. The trend is similar to what we observed before: best-of-$N$ plateaus after a certain number of LLM calls, while our approach scales as the compute budget increases, delivering an improvement in performance for each newly included AuPair. The self-repair baseline performs better

---

**Algorithm 3** Pair Generation

**Require:**
$\begin{cases} \text{LLM} & \text{large language model} \\ \mathcal{D}_{\text{train}} & \text{training dataset} \\ k & \text{number of few-shot examples} \\ N & \text{total number of LLM calls} \\ \text{score} & \text{code eval function} \end{cases}$

1: init candidate pairs $\mathcal{C} \leftarrow \{\}$
2: **for** $i = 1, \ldots, N$ **do**
3:      sample problem from dataset: $x \sim \mathcal{D}_{\text{train}}$
4:      sample $k$ pairs to use in-context: $c_1, \ldots, c_k \sim \mathcal{C}$
5:      build $k$-shot prompt: $p \leftarrow c_1 \parallel \ldots \parallel c_k \parallel x$
6:      generate fix: $\hat{y} \leftarrow \text{LLM}(p)$
7:      evaluate fix: $s_{\hat{y}} \leftarrow \text{score}(\hat{y})$
8:      **if** $s_{\hat{y}} > s_x$ **then**
9:          create new pair: $c \leftarrow \langle x, \hat{y} \rangle$
10:         add to candidate pairs: $\mathcal{C} \leftarrow \mathcal{C} \cup c$
11:         **if** $s_{\hat{y}} < 1$ **then**
12:             create new problem $\hat{x}$ with guess $\hat{y}$
13:             add new problem to dataset: $\mathcal{D}_{\text{train}} \leftarrow \mathcal{D}_{\text{train}} \cup \hat{x}$
14:         **else**
15:             remove all instances of problem from dataset: $\mathcal{D}_{\text{train}} \leftarrow \mathcal{D}_{\text{train}} - \{x\}$
16:         **end if**
17:      **end if**
18: **end for**
     **return** $\mathcal{C}$

---

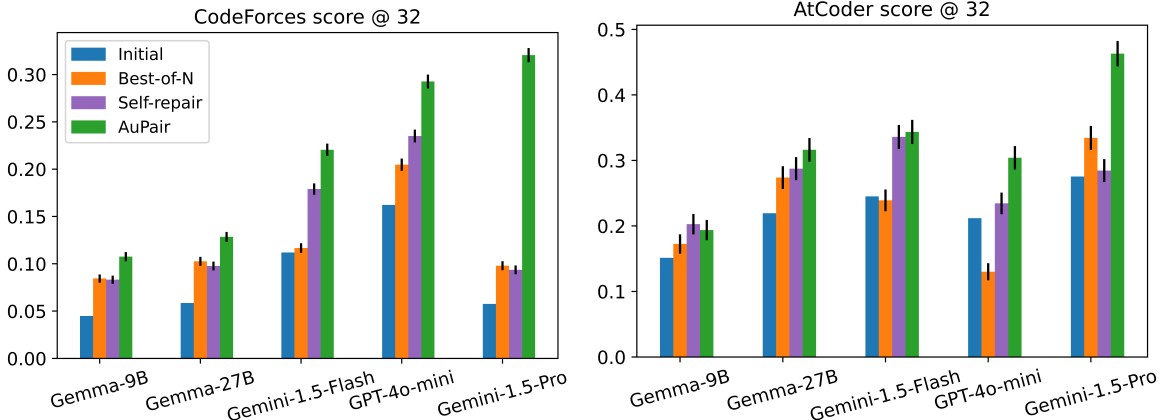

Figure 11: **In-distribution code repair performance for the strict accuracy metric** with $N = 32$ LLM calls at inference time. CodeForces (left) and AtCoder (right).

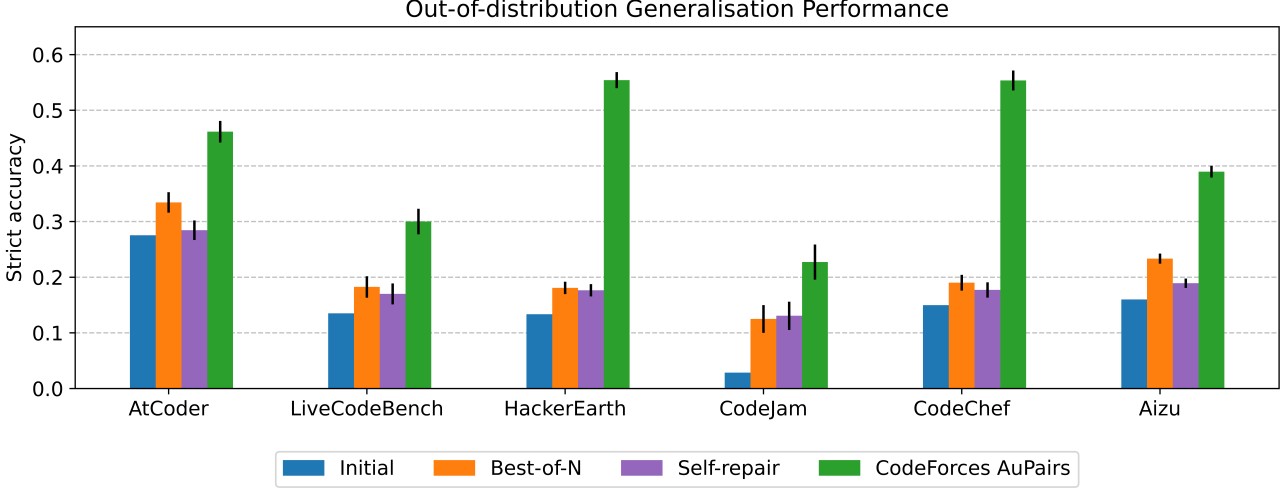

Figure 12: **Out-of-distribution code repair performance for the strict accuracy metric:** AuPairs extracted on the CodeForces dataset show strong generalisation performance across the other six datasets (the above results are obtained using Gemini-1.5-Pro)

with the Gemini-1.5-Flash model than with the Pro model; our hypothesis is that since the initial guesses for the Pro model were worse because of formatting issues, self-repair did not yield significant improvements. However, when the initial guesses are better, the self-repair baseline shows a stronger scaling result. Our algorithm yields 110 AuPairs and achieves a final test pass rate of 37.83% and strict accuracy 24.14%. Best-of-$N$, on the other hand, given the same budget of 110 LLM calls, has a test pass rate of 21.8% and strict accuracy 11.93%. Self-repair with the same compute budget has a final test pass rate of 34.1% and strict accuracy 22.39%. Since our AuPairs are selected submodularly, the initial pairs yield high returns in performance and these returns start diminishing slowly, but notably, performance does not plateau yet. Thus, it is abundantly clear that using AuPairs has a distinct advantage over currently used approaches like best-of-$N$ and self-repair in improving performance at inference time as compute budget increases.

### A.4. Cross-model Transfer to GPT-4o

In this section, we study the ability of the model to generalise to GPT-4o. We report results using AuPairs collected on GPT-4o-mini, since it is a weaker model than GPT-4o but from the same model family, and Gemini-1.5-Pro, since it is the strongest model from a different model family. The results in Table 5 validate our previous finding that in spite of using AuPairs from other models, there are significant performance gains (11% and 20% absolute performance gains) over the strongest baseline.

| Approach | Test pass rate |
|---|---|
| Initial | 0.244 |
| Best-of-$N$ | 0.100 |
| Self-repair | 0.374 |
| Gemini-1.5-Pro AuPairs | **0.486** |
| GPT-4o-mini AuPairs | **0.573** |

Table 5: **Transfer to GPT-4o:** We see strong cross-model transfer using GPT-4o-mini and Gemini-1.5-Pro AuPairs with GPT-4o, with 11% and 20% absolute performance improvement respectively over the strongest baseline.

### A.5. AuPair v/s RAG

Our approach yields a fixed ordered set of examples, of which the first $N$ examples are used at inference time, depending on the inference compute budget $N$. Retrieval-Augmented Generation (RAG) techniques (Lewis et al., 2021; Gao et al., 2024), on the other hand, retrieve examples that are in closest proximity to a particular test problem in the embedding space. As a result, for each test problem, a different set of $N$ examples is retrieved. Table 6 shows a comparison between RAG and AuPair for an inference compute budget $N = 32$ on the CodeForces dataset. The RAG baseline retrieves the top $N$ examples from the candidate set of *all* pairs (cardinality of each set of candidate pairs per model is given in Table 4). We use BERT (Devlin et al., 2019) embeddings to retrieve the top $N$ pairs.

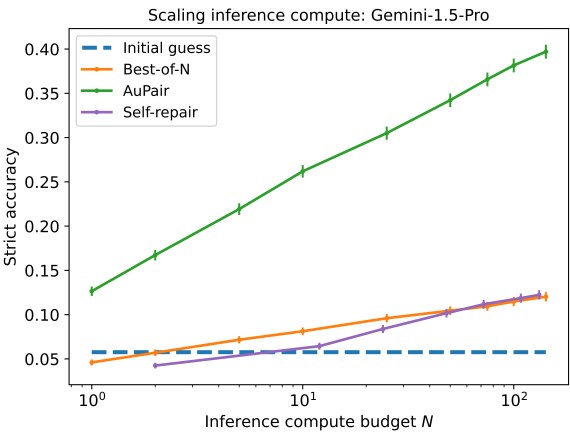 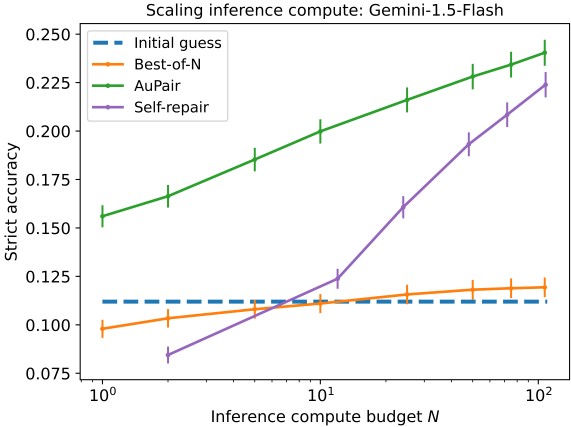

Figure 13: **Strict accuracy when scaling inference-time compute**: with $N = 144$ for Gemini-1.5-Pro and $N = 110$ for Gemini-1.5-Flash

| Model | RAG score | AuPair score |
|---|---|---|
| Gemini-1.5-Pro | 0.379 | **0.438** |
| GPT-4o-mini | 0.361 | **0.378** |
| Gemini-1.5-Flash | 0.318 | **0.352** |
| Gemma-27B | 0.178 | **0.214** |
| Gemma-9B | 0.156 | **0.198** |

Table 6: **Comparison with RAG:** We do an apples-to-apples comparison between AuPair and RAG on the Code-Forces dataset, and the results conclusively show that AuPair outperforms RAG across models.

### A.6. Code Repair with LiveCodeBench

Generalisation of AuPair prompting is important to improve code repair of smaller datasets. We posit that the AuPairs contain diverse code changes that transfer meaningfully across datasets, which may be important to those with scarce data, since out-of-distribution generalisation becomes especially relevant when we have small datasets, on which it can be quite difficult to obtain many different AuPairs.

We now show some the results obtained for a smaller dataset (400 problems) LiveCodeBench (LCB) (Jain et al., 2024). We generate the same train/val/test split ($37.5/12.5/50\%$) over 400 problems and apply our AuPair approach to obtain in distribution AuPairs for LCB.

Fig. 14 shows that even with smaller number of selected AuPairs we still obtain a gain over best-of-$N$ prompting. We obtained 5 AuPairs with the submodular extraction in Algorithm 2 for all the models except Gemma-9B which obtained only 3 AuPairs. The results indicate that performance with CodeForces AuPairs matches or exceeds that of using in-distribution AuPairs from LiveCodeBench. We also observe similar performance for in-distribution AtCoder

AuPairs and out-of-distribution CodeForces AuPairs on the AtCoder dataset.

Another interesting result in Fig. 14 is that both metrics, the test pass rate and strict accuracy, are comparable when using in-distribution AuPairs from LiveCodeBench and out-of-distribution AuPairs from CodeForces. This reinforces the insight mentioned earlier that extracting AuPairs on one dataset could lead to significant improvements over baselines even on other datasets.

### A.7. Coverage of Problem Categories is Preserved

The CodeForces dataset is richly annotated with category labels for each problem. A problem may have multiple tags, for instance, `strings` and `two pointers`. We use these fine-grained tags to study how the problem distribution is affected by Phase 1 and Phase 2 of our method separately. Fig. 16 shows the proportions of these categories observed in the initial dataset, the full set of pairs generated during Phase 1, and the final AuPairs. The high-level result is encouraging, namely that the starting diversity is approximately preserved. Phase 1 yields pairs for every single category, even those that lie at the tail. Furthermore, the (sparser) distribution over categories for the AuPairs after Phase 2 still shows several problems from rare categories. This additional result consolidates our insight that AuPairs are highly diverse in the types of problems they contain.

### A.8. Lineage

Here we look at the lineage of each pair generated during phase 1 of our algorithm, pair generation. The key idea here is to see if the set of all pairs collected during the pair generation phase are *deeper* i.e., they generate iteratively better solutions for a smaller set of problems, or *broader* i.e., they generate solutions for a larger set of problems but

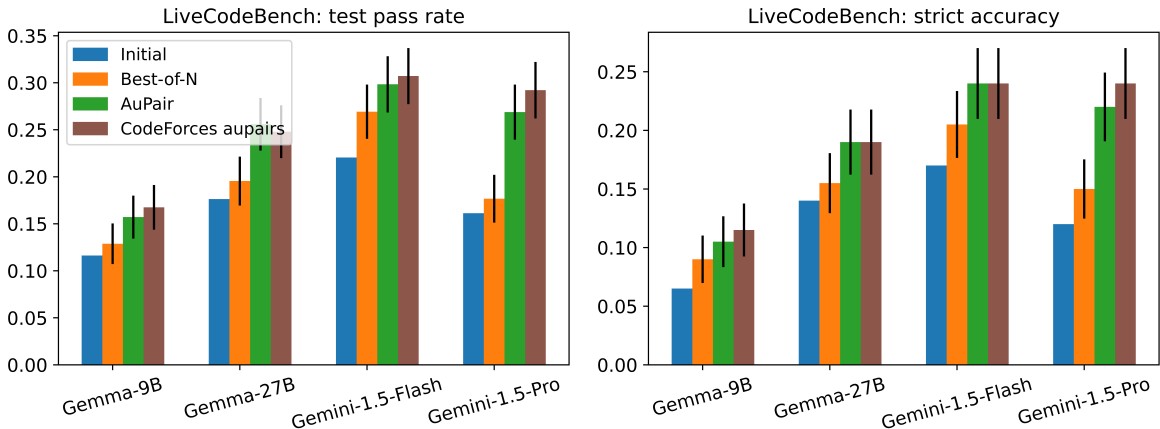

Figure 14: **LiveCodeBench results**: using AuPairs from the CodeForces dataset matches or outperforms in-distribution AuPairs from LiveCodeBench (left: test pass rate, right: strict accuracy).

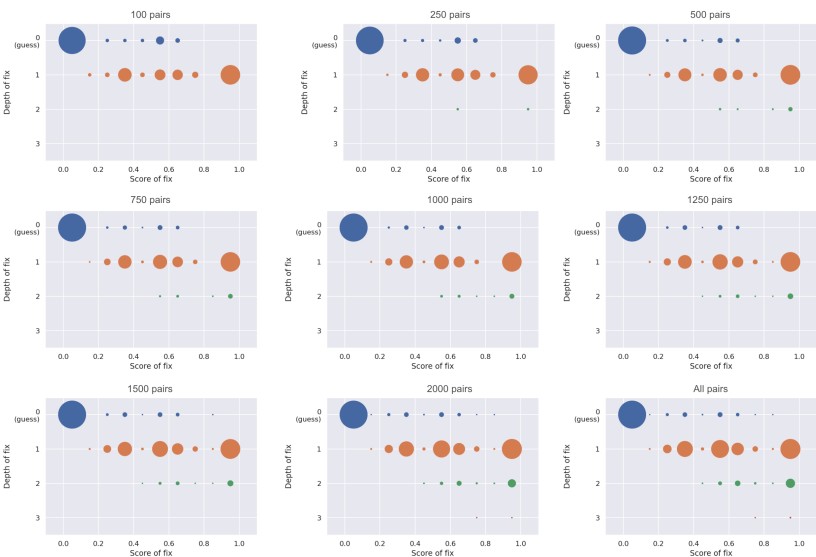

Figure 15: Visualising the lineage of the set of all pairs as the first phase of the algorithm, pair generation, progresses.

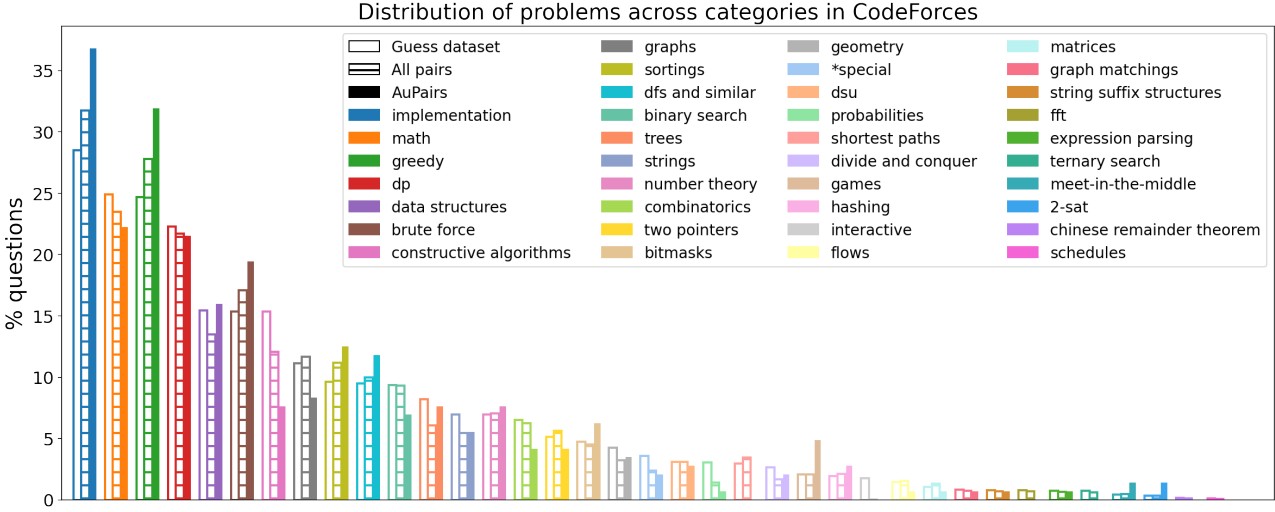

Figure 16: **Category-wise analysis:** analysing the distribution of AuPairs across different categories and comparing it with the distribution of problems in the dataset.

those solutions may not necessarily be perfect. The last plot in Fig. 15 (pairs generated on the CodeForces dataset using Gemini-Pro-1.5) indicates that the pairs collected have shallow lineage: a large proportion of guesses that had a score of 0 had corresponding fixes with perfect score at depth 1. We also see that the number of fixes decreases as depth increases (as seen from the size of the circles), indicating that several problems could not be improved beyond a certain point, or that they were not resampled during the pair generation phase. In both these cases, one solution is to allow more LLM calls during phase 1 to allow each problem to be sampled for repair more times. The takeaway here is that more sophisticated fixes for difficult problems can be discovered as we increase the budget of LLM calls during the pair generation phase. The entire evolution of this lineage at different points during pair generation is illustrated in Fig. 15.

### A.9. Code Diversity

We compute the code diversity score in Fig. 7(b) based on the number of different abstract syntax subtrees that each code instance produces. Algorithm 4 describes how this diversity score is computed. As a first step, for each guess in the test dataset and its corresponding fix generated by the LLM, we compute the respective abstract syntax subtrees. Next, we compute their corresponding set difference to get the unique subtrees for each code diff. This is done $N$ times for a compute budget of $N$ and the number of code diff subtrees across pairs and problems is averaged and normalised in the following manner to yield the diversity score $\delta$:

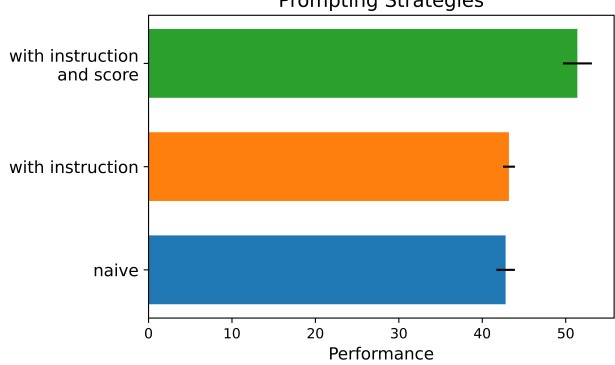

Figure 17: Ablations over repair prompt

$$\delta = \frac{1}{N|\mathcal{D}_{\text{test}}||S_{\max}|} \sum_{i=1}^{|\mathcal{D}_{\text{test}}|} |S_i^{\text{diff}}| \qquad (2)$$

where $S_i^{\text{diff}}$ is the set of all code diff subtrees generated with compute budget $N$ for problem $i$, and the normalising factor $S_{\max}$ corresponds to the highest number of subtrees that are present in any such set of code diff subtrees.

### A.10. Prompting

There are 2 types of prompts that we use: 1) guess generation prompt, and 2) repair prompt. The guess generation prompt is used during dataset creation, for obtaining the initial guesses for all problems in the dataset. The repair prompt is used throughout the rest of the paper: in the Pair

---

**Algorithm 4** Diversity score computation

**Require:** $\begin{cases} N & \text{inference compute budget} \\ \mathcal{D}_{\text{test}} & \text{test dataset} \\ \hat{Y} & \text{fixes for test problems} \\ f_{\text{AST}} & \text{abstract syntax subtree computation function} \end{cases}$

1: init set of code diff subtrees: $S^{\text{diff}} \leftarrow []$
2: **for** problem $x \in \mathcal{D}_{\text{test}}$ and its corresponding fixes $\hat{y} \in \hat{\mathcal{Y}}$ **do**
3:      compute abstract syntax subtrees for guess: $S^{\text{guess}} \leftarrow f_{\text{AST}}(x^{\text{guess}})$
4:      init code diff subtrees for this problem: $s \leftarrow \emptyset$
5:      **for** $j \in \{1, \ldots, N\}$ **do**
6:          compute abstract syntax subtrees for fix: $S_j^{\text{fix}} \leftarrow f_{\text{AST}}(\hat{y}_j)$
7:          update code diff subtrees: $s \leftarrow s \cup \{S_j^{\text{fix}} \setminus S^{\text{guess}}\}$
8:      **end for**
9:      append to code diff subtrees: $S^{\text{diff}} \leftarrow S^{\text{diff}} + s$
10: **end for**
11: compute normalising factor: $S_{\max} \leftarrow \text{argmax}_i S_i^{\text{diff}}$
12: compute diversity score: $\delta \leftarrow \frac{1}{N|\mathcal{D}_{\text{test}}||S^{\max}|} \sum\limits_{i=1}^{|\mathcal{D}_{\text{test}}|} |S_i^{\text{diff}}|$

**return** $\delta$

---

Generation (Phase 1, §2.1 with $k = 32$ random examples) and in the AuPair Extraction (Phase 2, §2.2) and during inference, with $k = 1$. The function signature indicates that the function expects a string as an input. The instruction specifies that the final answer is meant to be printed *inside* the function, and that the main function is not meant to be written.

The structure of our repair prompt is as follows: there is an instruction at the top, followed by the few-shot examples in the format: question, guess, fix. We also add the score achieved by the guess and the fix for the in-context example pairs. Following this, we add the text and initial guess for the problem and the LLM then has to generate a better fix. Note that we do not provide any extra execution feedback in the form of execution traces; this could potentially be explored by future work.

Our aim is clear: the pairs indicate a certain type of change and we provide these pairs in context to aid the LLM in generating an improved solution for the given problem. Some different prompting strategies that we tried out were the following:

*Naïve prompting*: only include the problem, guess and fix for the pairs, followed by the problem and guess for the test problem.

*Prompting with instruction only*: include the header instruction followed by the components of the naïve prompting strategy.

*Prompting with instruction and score*: include the elements of 2 above, but in addition, also include the score that each guess and fix received on the corresponding problem's test

cases. This is the prompt that we finally use and the one that gives us better results when compared using the same set of pairs with the previous 2 strategies. An important thing to note here is that we prompt the model with a desired fix score of 100 for the test problem.

We test the three strategies described above on a subset of the CodeForces dataset and report their performance in terms of number of problems solved, in the figure on the right. The results clearly indicate that the final prompting strategy that includes the instruction and score is the best strategy and so we choose it to compose the repair prompt.

### A.11. Code Execution

When the LLM generates a fix for any problem, we call the `solve()` function for each test case associated with that problem. We then compare the output with the ground truth and give a partial score corresponding to the proportion of test cases passed by this fix.

An important point to note is that the `solve()` function has to take as input a string, which is then parsed into the correct variables. This formatting requirement is a key reason for the poor initial performance of Gemini-1.5-Pro in Fig. 4. Since the instruction for generating the initial guess is not correctly followed by the model, a lot of guesses end up invariably having incorrect parsing of the input, leading to low scores. A lot of AuPairs extracted using these models, as a result, contain this formatting fix, as we will see in Section A.12.

---

**Guess Generation Prompt**

```
<problem text>
```
Complete the function definition below. Print the final answer in the function. Do not write main. Do not write anything outside the `solve()` function.

```
def solve(s: str):
    ...
```

---

**Repair Prompt**

You are an experienced software developer.
Look at the question (Q) and solutions below (A).
The main objective is to improve the `solve()` function to answer the question.

Example 1:

(Q): ...
Bad solution code `A(bad)`:

```
def solve(s: str):
    ...
```

The score of this code is `score(A(bad))` = `<example_guess_score>`.

Good solution code `A(good)`:
The score of this code is `score(A(good))` = `<example_fix_score>`.

```
def solve(s: str):
    ...
```

⋮

```
=====================================
```

The main objective is to improve the `solve()` function to answer the question.
(Q): ...
Bad solution code `A(bad)`:

```
def solve(s: str):
    ...
```

The score of this solution is `score(A(bad))` = `<guess_score>`

Good solution code `A(good)`:
The score of this solution is `score(A(good))` = `100`

Figure 18: Composing the repair prompt.

## A.12. Types of Fixes in AuPairs

We now show some examples of AuPairs and highlight the differences between the guess and fix for each pair. These are a mix of CodeForces pairs collected using different models. The scores achieved by the guess and fix on the corresponding problem's test cases are specified at the top right corner for each example in Fig. 18. We also provide a short description for each type of fix in the caption for ease of understanding. The types of pairs discovered using our algorithm cover a large area of potential fixes that can be made to an initial buggy piece of code: from smaller ones like parsing, fixing logical bugs pertaining to indexing errors, variable initialisations, etc., to larger changes like rewriting parts of the code, or even suggesting alternate routes to solve the same problem.

```
score: 0.33

def solve(n: str):
  n = int(n)
  dp = [0] * (n + 1)
  dp[0] = 1
  for i in range(1, n + 1):
    dp[i] = dp[i - 1] * 4
  print(dp[n])
```

```
score: 0.67

def solve(n: str):
  n = int(n)
  dp = [[0 for _ in range(4)] for _ in range(n + 1)]
  dp[0][0] = 1
  for i in range(1, n + 1):
    for j in range(4):
      for k in range(j, 4):
        dp[i][k] = (dp[i][k] + dp[i - 1][j]) %
1000000007
  print(sum(dp[n]))
```

Fix: converts a 1-D dynamic programming solution to 2-D.

```
score: 0.0

def solve(s: str, t: str, k: int):
  n = len(s)
  m = len(t)
  count = 0
  for i in range(n - m + 1):
    match = True
    for j in range(m):
      if abs(i + j - s.find(t[j], i, i + m)) > k:
        match = False
        break
    if match:
      count += 1
  print(count)
```

```
score: 1.0

def solve(s: str):
  n, m, k = map(int, s.split('\n')[0].split())
  S = s.split('\n')[1]
  T = s.split('\n')[2]
  count = 0
  for i in range(n - m + 1):
    match = True
    for j in range(m):
      found = False
      for p in range(max(1, i + j - k), min(n, i + j
+ k) + 1):
        if S[p - 1] == T[j]:
          found = True
          break
      if not found:
        match = False
        break
    if match:
      count += 1
  print(count)
```

Fix: composition of 1) input parsing correction, and 2) logical bug fix.

```
score: 0.5

def solve(s: str):
  n, a, b, c = map(int, s.split())
  if a <= b:
    print(n // a)
  else:
    glass_bottles = n // b
    remaining_money = n - glass_bottles * b
    plastic_bottles = remaining_money // a
    print(glass_bottles + plastic_bottles)
```

```
score: 1.0

def solve(s: str):
  n, a, b, c = map(int, s.split())
  if a <= b:
    print(n // a)
  else:
    glass_bottles = n // b
    remaining_money = n - glass_bottles * b
    plastic_bottles = remaining_money // a
    if remaining_money >= b - c:
      glass_bottles += 1
      remaining_money -= b - c
      plastic_bottles = remaining_money // a
    print(glass_bottles + plastic_bottles)
```

Fix: add an extra condition for edge cases.

```
score: 0.5
def solve(s: str):
  n = int(s.split('\n')[0])
  a = list(map(int, s.split('\n')[1].split()))
  count = 0
  for i in range(n):
    sum = 0
    for j in range(i, n):
      sum += a[j]
      if sum != 0:
        count += 1
  print(count)
```

```
score: 1.0
def solve(s: str):
  n = int(s.split('\n')[0])
  a = list(map(int, s.split('\n')[1].split()))
  count = 0
  for i in range(n):
    sum = 0
    for j in range(i, n):
      sum += a[j]
      if sum != 0:
        count += 1
      else:
        break
  print(count)
```

Fix: add loop exit condition.

```
score: 0.0
def solve(s: str):
    n = int(input())
    a = list(map(int, input().split()))
    if sum(a) == 0:
        print("NO")
        return
    print("YES")
    if a[0] != 0:
        print(1)
        print(1, n)
    else:
        print(n)
        for i in range(1, n + 1):
            print(i, i)
```

```
score: 0.25
def solve(s: str):
    n = int(s.split('\n')[0])
    a = list(map(int, s.split('\n')[1].split()))
    if sum(a) == 0:
        print("NO")
    else:
        print("YES")
        print(n)
        for i in range(n):
            print(i + 1, i + 1)
```

Fix: composition of 1) input parsing correction, and 2) logical bug fix.

```
score: 0.0
def solve(n: int, d: int, a: list[int]):
    balance = 0
    bank_visits = 0
    for i in range(n):
        balance += a[i]
        if balance < 0:
            if balance + d < 0:
                print(-1)
                return
            bank_visits += 1
            balance = d + balance
        if balance > d:
            print(-1)
            return
    print(bank_visits)
if __name__ == "__main__":
    n, d = map(int, input().split())
    a = list(map(int, input().split()))
    solve(n, d, a)
```

```
score: 0.67
def solve(s: str):
    n, d = map(int, s.split('\n')[0].split())
    a = list(map(int, s.split('\n')[1].split()))
    balance = 0
    bank_visits = 0
    for i in range(n):
        balance += a[i]
        if balance > d:
            print(-1)
            return
        if a[i] == 0 and balance < 0:
            bank_visits += 1
            balance = d
    print(bank_visits)
```

Fix: composition of 1) function signature correction, 2) input parsing correction, and 3) logical bug fix.

score: 0.0

```python
def solve(s: str):
    n = int(input())
    s = input().split()
    p = [int(x) for x in s]
    ans = 1
    for i in range(n):
        ans = ans * p[i] % (10**9 + 7)
    print(ans)
```

score: 0.5

```python
def solve(s: str):
    n = int(s.split('\n')[0])
    p = list(map(int, s.split('\n')[1].split()))
    ans = 1
    for i in range(n):
        ans = ans * p[i] % (10**9 + 7)
    print(ans)
```

Fix: input parsing correction.

score: 0.2

```python
def solve(s: str):
  n, p = map(int, s.split())
  dp = [float('inf')] * (n + 1)
  dp[0] = 0
  for i in range(1, n + 1):
    for j in range(32):
      if (1 << j) <= i:
        dp[i] = min(dp[i], dp[i - (1 << j)] + 1)
    if i - p >= 0:
      dp[i] = min(dp[i], dp[i - p] + 1)
  if dp[n] == float('inf'):
    print(-1)
  else:
    print(dp[n])
```

score: 0.6

```python
def solve(s: str):
  n, p = map(int, s.split())
  cnt = 0
  while n > 0:
    if n & 1:
      cnt += 1
    n >>= 1
  if p != 0:
    cnt += (n + p - 1) // p
  print(cnt)
```

Fix: solve problem using bit manipulation instead of dynamic programming.

score: 0.0

```python
def solve(s: str):
  n = int(s.split('\n')[0])
  graph = [[] for _ in range(n + 1)]
  for i in range(1, n):
    u, v = map(int, s.split('\n')[i].split())
    graph[u].append(v)
  visited = [False] * (n + 1)
  count = 0
  for i in range(1, n + 1):
    if not visited[i]:
      count += 1
      dfs(i, graph, visited)
  print(count - 1)
  res = []
  for i in range(1, n + 1):
    if not visited[i]:
      res.append(i)
  print(*res)
def dfs(node, graph, visited):
  visited[node] = True
  for neighbor in graph[node]:
    if not visited[neighbor]:
      dfs(neighbor, graph, visited)
```

score: 0.5

```python
def solve(s: str):
  n = int(s.split('\n')[0])
  graph = [[] for _ in range(n + 1)]
  for i in range(1, n):
    u, v = map(int, s.split('\n')[i].split())
    graph[u].append(v)
  def dfs(node, visited):
    visited[node] = True
    for neighbor in graph[node]:
      if not visited[neighbor]:
        dfs(neighbor, visited)
  count = 0
  visited = [False] * (n + 1)
  for i in range(1, n + 1):
    if not visited[i]:
      count += 1
      dfs(i, visited)
  print(count - 1)
  res = []
  visited = [False] * (n + 1)
  for i in range(1, n + 1):
    if not visited[i]:
      dfs(i, visited)
      res.append(i)
  print(*res)
```

Fix: partial correction to depth-first search graph algorithm.

```
score: 0.0
def solve(s: str):
  n, p, k = map(int, s.split()[0:3])
  a = list(map(int, s.split()[3:3+n]))
  s = [list(map(int,
s.split()[3+n+i*p:3+n+(i+1)*p])) for i in range(n)]
  people = sorted(enumerate(a), key=lambda x: x[1],
reverse=True)
  max_strength = 0
  for i, (person_index, audience_strength) in
enumerate(people):
    if i == k:
      break
    max_strength += audience_strength
    for j in range(p):
      max_strength_for_position =
max(max_strength_for_position, s[person_index][j])
    max_strength += max_strength_for_position
  print(max_strength)
```

```
score: 1.0
def solve(s: str):
  n, p, k = map(int, s.split()[0:3])
  a = list(map(int, s.split()[3:3+n]))
  s = [list(map(int,
s.split()[3+n+i*p:3+n+(i+1)*p])) for i in range(n)]
  people = sorted(enumerate(a), key=lambda x: x[1],
reverse=True)
  max_strength = 0
  for i in range(k):
    person_index = people[i][0]
    max_strength += a[person_index]
  for j in range(p):
    best_player_index = -1
    best_player_strength = -1
    for i in range(n):
      if i not in [person[0] for person in
people[:k]]:
        if best_player_strength < s[i][j]:
          best_player_strength = s[i][j]
          best_player_index = i
    max_strength += best_player_strength
  print(max_strength)
```

Fix: rewrite partial solution to pass all test cases.

Figure 18: Examples of AuPairs produced by our algorithm (multiple models represented above)

