# OpenReview forum: "AuPair: Golden Example Pairs for Code Repair"
_ICML.cc/2025/Conference — ICML 2025 poster_

### Official Review · Reviewer_2Ysj · 2025-03-13

**Overall Recommendation:** 3

**Summary:**

The paper introduces Aupair, which is a customed method that generates golden example pairs for enhancing code repair performance. In their work, each pair contains an initial guess and its fixes, which are used as in-context example at inference time to generate a repaired solution. During inference call, the fix with best score are selected as the output. The authors claim that this approach not only greatly outperforms methods like best‐of-N and self-repair on 5 LLMs and 7 datasets, but also scales better with on inference-time compute.

**Claims And Evidence:**

Claim: Aupair significantly improves code repair performance on different models and datasets, as well as scaling.
Evidence: Via experiments on the 7 datasets and the 5 LLMs, where the aupair shows better performance on different comparing metrics.

**Essential References Not Discussed:**

No.
However, regarding the code generations, it might be good if the authors can discuss related works that uses adversarial approaches for reference.

**Experimental Designs Or Analyses:**

The design of experiments seems reasonable to me, as it contains a good validation frame and across different LLMs. However, a more detailed discussion of parameters selection could benefit the paper more.

**Methods And Evaluation Criteria:**

The method is about generating guess-fix pairs and select a diverse subset by using their method. The evaluation seems good as it covers repair quality as well as diversity. So it make sense to me.

**Other Comments Or Suggestions:**

Consider include a more comprehensive ablation studies so we know the impacts with more details.
Consider discuss potential avoid-situation that may cause failures or lower performance.
Main concern: You didn't mention about the code availability, please ensure to make them available for reproducibility.

**Other Strengths And Weaknesses:**

Strong empirical results based on the datasets and LLMs the author presented.

Weakness:
Only unit test scores as the metric to evaluate correctness of code repairs.
Does not has good analysis on failure cases or situation that the method not working well
Limited theoretical part

**Questions For Authors:**

Have you tested the sensitivity of the input? Is the unit score the only metrics you used? have you considered better assessment of code quality?

**Relation To Broader Scientific Literature:**

The paper's Aupair is based on best-of-N, self-repair, and in-context learning. The main contribution seems to be a prompt-engineering method to improve the LLM's performance

**Theoretical Claims:**

There seems no theoretical proof.

---

> ### Author Rebuttal · Authors · 2025-03-31
>
> 1. discuss related work that uses adversarial approaches for reference.
>
> We will include references to adversarial approaches such as [1,2] in the Related Work section.
>
> 2. Only unit test scores as the metric to evaluate correctness of code repairs
>
> In this work, we have used unit test scores for assessment of fix quality and for guiding the exploration phase, such as has been done in prior work [3,4,5]. However, there is nothing fundamentally in the approach that prevents the use of other feedback mechanisms, such as reward models, or other sources of feedback to boost its performance. We leave this set of experiments for future work since we believe it is orthogonal to the key ideas presented.
>
> 3. Does not has good analysis on failure cases or situation that the method not working well
>
> We refer the reviewer to point 1 in our rebuttal address to reviewer oGUo: we provide concrete failure modes in which using AuPairs can lead to fixes that are potentially worse than the initial guesses, and this is observed more frequently for AuPair than other baselines on average, since AuPair is a diversity-boosting algorithm.
>
> 4. Is the unit score the only metrics you used?
>
> No, in addition to the test pass rate for unit tests, we also report the strict accuracy metric [6], which is the percentage of test problems that are fully solved. We report this metric for every single ablation in the paper; we refer the reviewer to Section A.2 in the Appendix, as well as Figs. 10, 11, 12 for plots showing this metric.
>
> 5. Consider discuss potential avoid-situation that may cause failures or lower performance.
>
> We refer the reviewer to Section A.4 of the Appendix, where we have a detailed discussion on the impact of smaller datasets on the efficacy of our proposed approach. Our results indicate that even though AuPair works well in the small dataset regime, curating the AuPairs on a larger dataset and then applying them to a smaller dataset, even though out-of-distribution, recovers in-distribution performance; it can also potentially lead to stronger scaling, since there will be fewer AuPairs extracted on smaller datasets. Furthermore, we have additional results for this in point 4 of our rebuttal to reviewer oGUo.
>
> ---
> [1] Adversarial patch generation for automated program repair, Alhefdhi et al. 2023
>
> [2] Learning to repair software vulnerabilities with Generative Adversarial Networks, Harer et al. 2018
>
> [3] Code Repair with LLMs gives an Exploration-Exploitation Tradeoff, Tang et al. 2024
>
> [4] Cycle: Learning to self-refine the code generation, Ding et al. 2024
>
> [5] Teaching LLMs to self-debug, Chen et al. 2023
>
> [6] Measuring coding challenge competence with apps, Hendrycks et al. 2021

---

### Official Review · Reviewer_oGUo · 2025-03-14

**Overall Recommendation:** 4

**Summary:**

The paper introduces AuPair, a novel algorithm designed to improve Large Language Models' (LLMs) performance on code repair tasks through inference-time computation. AuPair leverages in-context learning by synthesizing an ordered set of example pairs (called "AuPairs") consisting of initially incorrect code ("guesses") and subsequent corrected code ("fixes"). At inference time, the method provides each AuPair as a one-shot in-context example to guide LLMs toward generating diverse and improved code fixes. The inference-time algorithm to construct highly effective "golden example pairs" for code repair can achieve significant performance boosts over traditional approaches like best-of-N and self-repair.

**Claims And Evidence:**

- The authors claim their proposed approach (AuPair) significantly improves the code-repair capability of Large Language Models (LLMs) by leveraging carefully selected golden example pairs for in-context prompting.
- AuPair demonstrates strong generalization capability
- AuPair significantly outperforms traditional inference-time methods like Best-of-N and Self-repair methods.
- AuPair scales notably better with increasing inference-time compute budgets, yielding substantially higher performance improvements per unit of compute than baselines.
- AuPairs result in more diverse code fixes compared to best-of-N approaches

The claims are well supported.

**Essential References Not Discussed:**

N/A

**Experimental Designs Or Analyses:**

The experimental designs and analyses are sound.

**Methods And Evaluation Criteria:**

AuPair consists of two phases:

Phase 1: Pair Generation
- Data Collection: Starting with a dataset of coding problems and initial LLM-generated guesses (potentially flawed code), the LLM iteratively generates fixes.
- Candidate Pair Generation: For each sampled problem and its initial flawed guess, the LLM produces improved fixes using a k-shot prompt (with randomly selected pairs from existing candidate set). Generated fixes that outperform the original guess are added to a pool of candidate guess-fix pairs. Imperfect fixes become new guesses to generate additional candidate pairs.

Phase 2: AuPair Extraction
- For each candidate pair and problem in the validation set, the algorithm constructs a 1-shot prompt to evaluate how effectively each pair guides the LLM to repair other problems.
- The LLM-generated fixes are scored against provided unit tests, forming a fix-quality matrix.
- AuPairs are selected in a greedy manner from the candidate pairs based on their incremental contribution to solving distinct problems, ensuring complementarity and diversity.
- This selection iteratively picks pairs that yield maximum additional performance on unsolved problems until further improvements fall below a predefined tolerance threshold.

The paper evaluates the proposed method using seven competitive programming datasets, using multiple models.

These methods and metrics allow for a rigorous evaluation of the algorithm's effectiveness.

**Other Comments Or Suggestions:**

N/A

**Other Strengths And Weaknesses:**

Strengths:
- The paper introduces a practical and impactful approach
- Extensive experiments across multiple competitive programming datasets (CodeForces, AtCoder, CodeChef, etc.)
- Empirical results convincingly show substantial improvements
- Detailed ablation and additional analysis

Weaknesses:
- Limited exploration of failure cases
- The evaluations focus predominantly on competitive programming tasks. It remains unclear whether the impressive results would translate directly into other code repair domains, such as debugging production software (or non-code related domain), where test coverage or complexity might differ significantly.
- The paper's method strongly relies on predefined test cases and evaluations on these tests to judge fix quality.

**Questions For Authors:**

How sensitive is AuPair's effectiveness to the size and diversity of the validation set used for AuPair selection?

**Relation To Broader Scientific Literature:**

The paper builds upon the broader literature which has shown that increasing inference-time computation, without additional fine-tuning, significantly enhances Large Language Model (LLM) performance (e.g., best-of-N, self-consistency, etc). The paper effectively integrates ideas from multiple recent lines of research—leveraging in-context learning capabilities, automated code repair without supervised fine-tuning, and strategic prompting—to deliver significant performance gains, and positions itself well within and advances the state of the literature in automated self-repair for code generation.

**Theoretical Claims:**

The paper does not make explicit theoretical claims or provide formal theoretical analysis

---

> ### Author Rebuttal · Authors · 2025-03-31
>
> 1. Limited exploration of failure cases
>
> While AuPairs have been shown to significantly boost performance, they can occasionally have unintended impacts as well. The following table contains the percentage of CodeForces problems in which some fixes had a decrease in fix score compared to the initial guess. Note that this does not affect any of the performance results in the main paper, since for measuring performance, the best scoring response is selected.
>
> **Model** | **Approach** | **% problems w/ decreased fix score**
> --- | --- | ---
> Gemini-1.5-Pro | Best-of-$N$ | 10.52
> &nbsp; | Self-repair | 7.62
> &nbsp; | AuPair | 11.63
> GPT-4o-mini | Best-of-$N$ | 20.09
> &nbsp; | Self-repair | 11.87
> &nbsp; | AuPair | 15.28
> Gemini-1.5-Flash | Best-of-$N$ | 9.47
> &nbsp; | Self-repair | 22.28 |
> &nbsp; | AuPair | 11.79
> Gemma-27B | Best-of-$N$ | 14.86
> &nbsp; | Self-repair | 9.72
> &nbsp; | AuPair | 15.21
> Gemma-9B | Best-of-$N$ | 13.16
> &nbsp; | Self-repair | 9.38
> &nbsp; | AuPair | 13.09
>
> As we can see from the above table, in most cases, using AuPair results in an increase in the number of problems for which a fix is worse than the initial guess. This is to be expected since AuPair is an algorithm that in addition to boosting performance also boosts diversity of the generated responses.
>
> 2. The evaluations focus predominantly on competitive programming tasks. It remains unclear whether the impressive results would translate directly into other code repair domains.
>
> Competitive programming tasks are a canonical domain for code repair for two reasons: these tasks are **rigorous** (can be precisely evaluated, cannot be gamed) and they are **hard** (supposed to differentiate between top human coders). Both of these make them appropriate to research and test new methods, the crisp, reliable evaluation of challenging tasks means that even differences of a few percent are meaningful capability improvements. But indeed, competitive programming tasks are not the bulk of what users might request. While we see no reason to doubt that our method will translate to broader, less well-defined, or easier tasks, we do not want to make a strong claim about this transferability; producing that evidence is out of scope for this paper. We will expand on this in the final version of the paper.
>
> 3. The paper's method strongly relies on predefined test cases and evaluations on these tests to judge fix quality.
>
> Indeed, in this work, we have used test cases for assessment of fix quality and for guiding the exploration phase, such as has been done in prior work [1,2,3]. However, contrary to other approaches for code repair, where failed test cases are given to the model in the prompt, we do not include test cases in the prompt; we use test cases solely to build the AuPairs.
>
> Furthermore, we would like to highlight that there is nothing fundamentally in the approach that prevents the use of other feedback mechanisms, such as reward models, or other sources of feedback to boost its performance. We leave this set of experiments for future work since we believe it is orthogonal to the key ideas presented.
>
> 4. How sensitive is AuPair's effectiveness to the size and diversity of the validation set used for AuPair selection?
>
> This is a great question; we conducted experiments with smaller validation sets to curate AuPairs and report the results below. We also include the results of the random baseline for calibration:
>
> Size of Validation | # of AuPairs | Score (inference budget = 32)
> -----------------------|------------------|-----------------------------------------
> Random	    | N/A	   | 0.383
> 10%		     | 32		   | 0.403
> 25%		     | 52		   | 0.418
> 100%		     | 144	   | 0.438
>
> The larger the validation set, the more distinct complementary improvements can be observed, and hence the larger the maximal set of AuPairs that can be discovered. So larger validation sets make it possible to effectively scale up to more inference compute. However, even just looking at just the top 32 AuPairs (which is apples-to-apples for varying validation set sizes), we find that their quality increases monotonically with the size of the validation set. We have also conducted additional analysis on the impact of smaller datasets, for which we point the reviewer to section A.4 in the paper.
>
> ---
>
> [1] Code Repair with LLMs gives an Exploration-Exploitation Tradeoff, Tang et al. 2024
>
> [2] Cycle: Learning to self-refine the code generation, Ding et al. 2024
>
> [3] Teaching LLMs to self-debug, Chen et al., 2023

---

### Official Review · Reviewer_g7mK · 2025-03-24

**Overall Recommendation:** 3

**Summary:**

Paper introduces AuPair, an inference time algorithm to improve code repair capabilities of LLM. The core idea lies in first a diverse of generating golden pairs (guess, fix) using an LLM and then using a submodular selection algorithm to identify and generate an ordered set of golden example pairs. During inference, these pairs are used sequentially as 1-shot in-context examples to guide the LLM. The paper illustrates extensive experiments across 5 LLMs and 7 code repair datasets, demonstrating that AuPair consistently and significantly outperforms best-of-N and self-repair

**Claims And Evidence:**

Yes.

**Essential References Not Discussed:**

1. Code Repair with LLMs gives an Exploration-Exploitation Tradeoff (Tang et al. NeurIPS 2024)
2. CYCLE: Learning to Self-Refine the Code Generation (Ding et al, OOPSLA 2024)

**Experimental Designs Or Analyses:**

Yes

**Methods And Evaluation Criteria:**

Yes.

**Other Comments Or Suggestions:**

See questions for authors

**Other Strengths And Weaknesses:**

Strengths:
1. Extensive experiments section covering various aspects of the approach (scaling, generalization, multiple datasets)
2. The approach is interesting for improving the program repair capabilities of LLMs without any fine-tuning/training.


Weakness:
1. The evaluation primarily focuses on models from the Gemini family and GPT-4o-mini. It omits other models recognized for strong coding performance (and high rankings on relevant benchmark leaderboards), such as Claude 3.5 Sonnet, GPT-4o, and DeepSeek-V3. Including a more diverse set of models would strengthen the evaluation.
2. All experiments are conducted within the domain of competitive programming. Checking if AuPair could be used for other programming tasks, such as programming-by-example (e.g., using datasets like those explored by Tang et al. in "Code Repair with LLMs Give an Exploration-Exploitation Trade-off"), would be valuable and help generalize the approach.
3. While the ablation study in Section 3.2 using random pairs provides insights into AuPair quality, further ablation experiments could help assess the effectiveness of the greedy selection approach. This could include comparing against randomly ordered AuPairs or exploring alternative selection criteria, such as a curriculum learning-inspired approach (e.g., selecting pairs in increasing order of problem difficulty).
4. Is there a reason why RAG over the candidates dataset (guess, fix) pairs is not considered as a baseline?

**Questions For Authors:**

1. Is it accurate to state that AuPair utilizes a slightly higher computational budget compared to best-of-N and self-repair baselines? AuPair involves LLM calls in two preprocessing stages (dataset creation and pair selection) before the budgeted N inference calls. The budget for the creation phase, ranging from 10,000 to 35,000 calls, is quite substantial.
2. The phrase "in conjunction with high performing" in line 260 is somewhat confusing, given that Gemini-1.5-Pro's initial performance is reported as low.
3. Section 2.2, describing the fix quality matrix, requires a more detailed explanation as it is central to the approach. The necessity of the subtraction step, in particular, is not intuitively clear and needs further justification.
4. Figure 7 suggests a performance drop when AuPairs are generated by a different model, even if the generating model is generally superior. More insights could help explain this phenomenon and the underlying reasons for this cross-model performance variation.?
5. Could there be another analysis on the diversity aspect which includes the type of repairs these pairs help in? For instance, incorrect formatting, syntax, semantic, different types of bugs?

**Relation To Broader Scientific Literature:**

While the paper focuses on code repair this can have impact on contributing to more robust and reliable software.

**Theoretical Claims:**

NA

---

> ### Author Rebuttal · Authors · 2025-03-31
>
> 1. Including a more diverse set of models would strengthen evaluation
>
> We have included 5 models spanning 3 model families across 7 datasets in the existing results – Gemini, GPT, Gemma; the results clearly indicate that AuPair works across models. We understand the reviewer's concern and agree that showing results on more models would strengthen evaluation, but getting the full set of results with another model within the short rebuttal period would not be feasible. What we can do, however, is include new transfer results with GPT-4o (suggested by reviewer) using AuPairs from GPT-4o-mini and Gemini-1.5-Pro.
>
> **Approach**|**Score on GPT-4o**
> ---|---
> Initial|0.244
> Best-of-$N$|0.100
> Self-repair|0.374
> w/ Gemini-1.5-Pro AuPairs|**0.486**
> w/ GPT-4o-mini AuPairs|**0.573**
>
> Even when using AuPairs from other models, we see **11% and 20% absolute performance gain** over baselines, thus solidifying our claim that even for more code-competent models such as GPT-4o, AuPair gives large performance gains.
>
> 2. RAG over the candidates dataset pairs is not considered as a baseline
>
> We implemented a RAG baseline, choosing the top 32 pairs for each problem, retrieved from the candidate pair dataset, as the reviewer suggested:
>
> Model|RAG score|AuPair score
> ---|---|---
> Gemini-1.5-Pro|0.379|**0.438**
> GPT-4o-mini|0.361|**0.378**
> Gemini-1.5-Flash|0.318|**0.352**
> Gemma-27B|0.178|**0.214**
> Gemma-9B|0.156|**0.198**
>
> We see that across all models, AuPair, which uses a fixed set of in-context examples, outperforms RAG, which requires the entire set of candidate pairs to be compared with each test problem to choose the in-context examples.
>
> 3. Another analysis on the diversity aspect which includes the type of repairs these pairs help in
>
> We show a breakdown of the repairs generated into 4 categories: 1) %problems with improved fix score, 2) %problems in which code was reformatted to obey constraints, 3) %problems in which fix was improved by just changing the logic, 4) %problems in which score remained unchanged. Composite changes (formatting + logical) are reported in formatting fixes.
>
> **Model**|**Approach**|**Improvements**|**Formatting fixes**|**Logical fixes**|**Unchanged**
> ---|---|---|---|---|---
> Gemini-1.5-Pro|Best-of-$N$|9.67|1.37|8.45|81.99
> &nbsp;|Self-repair|8.50|0.83|7.91|82.48
> &nbsp;|AuPair|47.14|50.32|14.11|44.82
> GPT-4o-mini|Best-of-$N$|8.66|2.48|7.39|61.92
> &nbsp;|Self-repair|12.98|2.71|10.81|60.24
> &nbsp;|AuPair|22.34|25.41|6.70|56.46
> Gemini-1.5-Flash|Best-of-$N$|8.27|0.05|8.24|82.92
> &nbsp;|Self-repair|18.52|0.12|18.52|54.77
> &nbsp;|AuPair|24.21|16.12|16.88|65.99
> Gemma-27B|Best-of-$N$|11.57|0.49|10.78|77.09
> &nbsp;|Self-repair|9.35|0.44|9.13|79.55
> &nbsp;|AuPair|18.97|13.90|11.32|69.24
> Gemma-9B|Best-of-$N$|16.76|7.48|12.77|72.66
> &nbsp;|Self-repair|10.60|0.42|10.50|79.85
> &nbsp;|AuPair|20.00|18.16|14.71|69.73
>
> Some insights:
>
> - Since Gemini-1.5-Pro guesses have more formatting bugs, % problems with formatting fixes using AuPair is high.
> - AuPair also helps the model repair solutions with logical errors.
> - AuPair yields responses that are more diverse than baselines in the test case scores, indicated by the lower value of "Unchanged".
>
> Example of composite fixes:
>
> Guess:
> ```
> def solve(s: str):
>    n = len(s)
>    a = int(input()) - 1
>    b = int(input()) - 1
>     cost = 0
>    for i in range(a, b):
>      if s[i] != s[i + 1]:
>        cost += 1
>    print(cost)
> ```
>
> Fix:
> ```
> def solve(s: str):
>    n, a, b = map(int, s.split('\n')[0].split())
>    companies = s.split('\n')[1]
>    cost = 0
>    if companies[a - 1] != companies[b - 1]:
>      cost = abs(a - b)
>    print(cost)
> ```
>
> 4. Does AuPair utilize a slightly higher computational budget compared to baselines?
>
> This is partially correct: there is a one-time amortised cost that our algorithm incurs to construct the AuPairs, but the cost at test-time is identical. Note that the same fixed set of AuPairs boosts performance across models and datasets (Fig. 6, 7 show out-of-distribution generalisation), indicating that the upfront cost is easily amortised.
>
> 5. The phrase "in conjunction with high performing" in line 260 is somewhat confusing
>
> We explain this in line 250: "since the code generated has to adhere to a specific format to allow successful execution, we observe that many initial guesses of generated code fail because they do not obey these conditions". Moreover, our experiments on GPT-4o above also show that the insight stands, since we observe absolute improvements of 11% and 20% using AuPair compared to self-repair (strongest baseline).
>
> 6. The necessity of the subtraction step is not intuitively clear
>
> The fix quality matrix contains the score for each candidate pair for each validation problem. After picking the pair with the best mean score, we subtract this score from the matrix because only after removing the previous best pair can we find the complementary next best pair.
>
> 7. Missing references
>
> We will include the mentioned references in the Related Work section.

---

### Decision · Program_Chairs · 2025-05-01

**Decision:**

Accept (poster)

**Comment:**

This paper presents an approach to synthesize an ordered set of AuPairs (golden example pairs) of an incorrect and subsequent corrected version of the code. The key idea of the approach is to select the AuPairs that are complementary to each other and maximize their usefulness based on a held-out validation set. These AuPairs are then used at inference-time for 1-shot in-context code repairs, and the results show that it yields a significant improvement over best-of-N and self-repair approaches. All reviewers found the overall idea of synthesizing an ordered set of golden example pairs (AuPairs) and then using them at inference time to compute diverse repair solutions  quite interesting. But there were also concerns around generalizability of the approach on other models, actual notion of diversity, RAG baseline, only using test-based metrics, failure analysis, and limited evaluation dataset of only competitive programming. The author response with additional experiments with GPT-4o, RAG baseline, further analysis of diversity and failure cases, and robustness of selection helped quite a lot. It would be great to include the additional results and discussions in the rebuttal response as well as incorporate feedback from the reviews in the final version of the paper.